# How Far Are We from Optimal Reasoning Efficiency?

**Jiaxuan Gao**[12]   **Shu Yan**[3]   **Qixin Tan**[1]   **Lu Yang**[1]
**Shusheng Xu**[2]   **Wei Fu**[12]   **Zhiyu Mei**[2]   **Kaifeng Lyu**[1]   **Yi Wu**[1*]
[1] IIIS, Tsinghua University   [2] Ant Group   [3] Nanjing University
{samjia2000, jxwuyi}@gmail.com

## Abstract

Large Reasoning Models (LRMs) demonstrate remarkable problem-solving capabilities through extended Chain-of-Thought (CoT) reasoning but often produce excessively verbose and redundant reasoning traces. This inefficiency incurs high inference costs and limits practical deployment. While existing fine-tuning methods aim to improve reasoning efficiency, assessing their efficiency gains remains challenging due to inconsistent evaluations. In this work, we introduce the *reasoning efficiency frontiers*, empirical upper bounds derived from fine-tuning a base LRM (DeepSeek-R1-Distill-Qwen-1.5B/7B) across diverse approaches and training configurations. Based on these frontiers, we propose the *Reasoning Efficiency Gap (REG)*, a unified metric quantifying deviations of any fine-tuned LRMs from these frontiers. Systematic evaluation on challenging mathematical benchmarks, AMC23, AIME24, and AIME25, reveals significant gaps in current methods: they either sacrifice accuracy for short length or use excessive tokens to achieve sub-optimal accuracies despite high overall accuracy. To reduce the efficiency gap, we propose REO-RL, a Reinforcement Learning algorithm that optimizes reasoning efficiency by targeting a sparse set of token budgets. Leveraging numerical integration over strategically selected budgets, REO-RL approximates the full efficiency objective with low error using a small set of token budgets. Experiments show that, compared to vanilla RL with outcome reward, REO-RL reduces the reasoning efficiency gap by 74.5% and 64.2% in the 1.5B and 7B settings. The 7B LRM fine-tuned with REO-RL achieves reasoning conciseness surpassing frontier LRMs like Qwen3 and Claude Sonnet 3.7. Ablation studies confirm the efficacy of our token budget strategy and highlight REO-RL's flexibility across design choices. This work establishes a systematic framework for evaluating and optimizing reasoning efficiency in LRMs. We will release the related code, data, and models to support future research on efficient reasoning in LRMs.

## 1   Introduction

Large Reasoning Models (LRMs) have recently emerged as a powerful class of models capable of solving complex tasks that require advanced reasoning. Frontier LRMs such as OpenAI o1 [OpenAI] and DeepSeek R1 [Guo et al., 2025] have obtained superior performance across a wide range of tasks, including mathematical reasoning and competitive programming. A major factor behind this success is their ability to perform deep, multi-step reasoning through extended Chain-of-Thought (CoT) processes. These reasoning traces often include sophisticated operations such as reflection, verification, and exploration, within a single inference pass.

However, the powerful capability of long CoT reasoning comes at a cost. LRMs frequently generate overly verbose and redundant reasoning traces, a phenomenon referred to as the *overthinking problem* [Yang et al., 2025b, Sui et al., 2025]. Recent studies [Chen et al., 2024b, Sui et al., 2025] have

---

[*] Corresponding author

shown that even simple questions like "2 + 3 = ?" can result in outputs spanning up to 900 tokens. This redundancy brings a significant cost in inference time and limits practical deployment. Several fine-tuning approaches have been proposed to improve reasoning efficiency, mostly focusing on reducing response length [Team et al., 2025, Luo et al., 2025a, Aggarwal and Welleck, 2025, Arora and Zanette, 2025, Yeo et al., 2025, Shen et al., 2025b, Qu et al., 2025, Yang et al., 2025a, She et al., 2025, Hou et al., 2025]. However, comparing these methods remains difficult due to inconsistent evaluation setups, including varying models, benchmarks, and mixed performance metrics. It is still unclear how close current approaches are to the optimal trade-off between length and accuracy.

In this work, we investigate a critical question: *How far are current approaches from reaching the optimal reasoning efficiency?* To answer this question, we conduct a comprehensive empirical study using two LRMs, DeepSeek-Distill-Qwen-1.5B and DeepSeek-Distill-Qwen-7B, on three challenging benchmarks, AMC23, AIME24, and AIME25. We introduce the concept of ***reasoning efficiency frontiers***, derived from fine-tuning the base LRMs with 3 types of algorithms and diverse training configurations. These reasoning efficiency frontiers represent the best reward achievable by the current approaches at each token budget, offering a practical lower bound on optimal efficiency. By comparing current methods to these frontiers, we uncover a substantial gap. Existing methods often fall short in one of two ways, either they aggressively shorten responses at the expense of accuracy, or methods that reach high overall accuracy would consume significantly more tokens than necessary to reach moderate accuracy levels. To quantify this gap, we propose the ***Reasoning Efficiency Gap (REG)***, a unified metric that captures both accuracy and response length by measuring the area between the length-accuracy curve of an LRM and the frontier. REG offers practical insights into how much room still remains for improvement.

We further ask: *How can an LRM be fine-tuned to minimize this efficiency gap?* A natural approach is to optimize the rewards across all possible token budgets at RL training time. However, this approach leads to a costly training process since rewards across all token budgets should be evaluated. To overcome the inefficiency of this dense reward approach, we introduce ***Reasoning Efficiency Optimization with Reinforcement Learning (REO-RL)***, a novel RL algorithm that improves reasoning efficiency by targeting a small set of token budgets. The key insight of REO-RL is that the total rewards across all token budgets can be well-approximated using numerical integration over a small set of representative token budgets. We experiment with both an oracle-based greedy selection based on the estimated reasoning efficiency frontiers and a heuristic strategy using exponentially spaced token budgets. Remarkably, with as few as 5 token budgets, both token selection strategies could achieve an approximation error of less than 1%, ensuring that the optimization direction of REO-RL could align with the total rewards across all token budgets.

Finally, through systematically evaluating the reasoning efficiency gap for existing methods and REO-RL, we find that REO-RL consistently outperforms baseline methods in terms of reasoning efficiency across model scales. Notably, compared to the vanilla RL baseline, REO-RL reduces the efficiency gap by 74.5% and 64.2% for the 1.5B and 7B models, respectively. By conducting a controlled comparison with frontier LRMs including Qwen3 and Claude Sonnet 3.7, we show that the 7B LRM fine-tuned with REO-RL exhibits more concise reasoning patterns than frontier LRMs. Our ablation study on the design choices for REO-RL reveals the success of the exponential token budget selection strategy and effective approximation with a small amount of token budgets. We also show that REO-RL is flexible and various design choices, including setting the coefficients uniformly as 1 and using question-specific token budget, both could lead to competitive reasoning efficiency improvements.

## 2 Related Works

**Efficient Reasoning.** Prior studies have shown that LRMs often suffer from redundant reasoning. Even for very simple questions, Frontier LRMs often generate lengthy responses spanning thousands of tokens [Chen et al., 2025, Sui et al., 2025]. This redundancy in the reasoning process brings significant overheads in the inference costs. Several works are then proposed to make LRM reasoning more concise. Team et al. [2025], Luo et al. [2025a], Aggarwal and Welleck [2025], Arora and Zanette [2025], Yeo et al. [2025], Shen et al. [2025b], Qu et al. [2025], Yang et al. [2025a], She et al. [2025], Hou et al. [2025] investigate RL training with length reward designs, mostly focusing on reducing the reasoning lengths. A line of works apply SFT to fine-tune LRMs on datasets with variable-length reasoning traces to elicit concise reasoning [Yu et al., 2024, Wang et al., 2023, Han et al., 2024] or

adjustable length control [Kang et al., 2024, Xia et al., 2025, Ma et al., 2025, Liu et al., 2024, Yu et al., 2025c]. Some works also investigate enhancing the reasoning efficiency through test-time techniques, including reward model guided decoding [Sun et al., 2024, Liao et al., 2025] and uncertainty-based dynamic reasoning [Fu et al., 2024, 2025], and confidence-based approaches [Taubenfeld et al., 2025, Huang et al., 2025]. In this work, we study the optimal reasoning efficiency for an LRM and focus on training-based approaches for enhancing LRM reasoning efficiency. Our method aims at enhancing the accuracy of the LRM under diverse token budgets without explicitly incentivizing shorter responses.

**RL for LRM Reasoning.**    Reinforcement Learning is the central technique for eliciting and enhancing the reasoning capability of LRMs. Frontier LRMs, including OpenAI o1 [OpenAI] and DeepSeek R1 [Guo et al., 2025], have shown that applying "zero RL" on a base LLM could effectively elicit the ability to utilize long CoTs for complex reasoning. A series of works have emerged with the focus on improving the training efficiency of RL for LRMs from the perspectives of data [Luo et al., 2025c, RL Lab, 2025, He et al., 2025, Li et al., 2025, Wang et al., 2025], algorithms [Guo et al., 2025, He et al., 2025, Luo et al., 2025c, Yu et al., 2025b, Yue et al., 2025], and training framework [Sheng et al., 2024, Luo et al., 2025b, RL Lab, 2025]. A number of works successfully apply zero RL training on a wide range of reasoning-heavy domains, including multi-modality [Shen et al., 2025a, Zhang et al., 2025], medical [Yu et al., 2025a, Chen et al., 2024a], and financial [Liu et al., 2025]. Recent works also explore efficiency enhancement by encouraging concise reasoning with RL [Team et al., 2025, Luo et al., 2025a, Aggarwal and Welleck, 2025, Arora and Zanette, 2025, Yeo et al., 2025, Shen et al., 2025b, Qu et al., 2025, Yang et al., 2025a, She et al., 2025, Hou et al., 2025]. In this work, we focus on enhancing the reasoning efficiency with RL.

# 3    Preliminary

**LRM Reasoning.**    In this work, we focus on the task of mathematical reasoning. Given a question $x$, the goal of an LRM policy is to generate a response $y$ that contains step-by-step reasoning to derive the correct answer. We assume access to a verifier $\mathcal{R}(x, y)$ that evaluates the correctness of a solution $y$ given the question $x$. In practice, such a verifier is implemented by matching the ground-truth answer and the model-generated answer. The LRM is a policy $\pi_\theta$ parameterized with $\theta$ and generates a sequence of reasoning tokens in an auto-regressive manner. Given a question distribution $\mathcal{D}$, the objective of the LRM is to maximize the probability of producing correct responses,

$$J(\mathcal{D}, \theta) = \mathbb{E}_{x \sim \mathcal{D}, y \sim \pi_\theta(\cdot|x)}[\mathcal{R}(x, y)] \tag{1}$$

where $\theta$ is in a parameter space $\Theta$ and the response length $|y|$ is limited to the maximum length $L_{\max}$. In practice, $\theta$ is usually obtained through applying fine-tuning approaches such as RL on a base LRM. Therefore we assume the existence of a base LRM $\theta_{\text{base}}$ and $\Theta$ to be the set of all LRMs that could be obtained by fine-tuning $\theta_{\text{base}}$ with any algorithm.

# 4    Understanding the Limits of Efficient Reasoning

## 4.1    Defining Optimality in Token-Bounded Reasoning

**Evaluating Reasoning under Token Budgets.**    To assess optimal reasoning efficiency, we must evaluate the performance of an LRM $\pi_\theta$ under a fixed token budget $L$. Simply truncating a model's output after $L$ tokens, however, may lead to incomplete responses. To address this, we define a fallback mechanism following prior works [Muennighoff et al., 2025, Fu et al., 2024]. If the reasoning trace $y \sim \pi_\theta(\cdot|x)$ exceeds $L$ tokens, the model is prompted to produce a final answer $a$ directly from the truncated trace $y_{:L}$,

$$a = \text{Answer}(\pi_\theta, x, y) = \begin{cases} \pi_\theta(\cdot|x, y_{:L}, [\text{The Final Answer is}]) & \text{if } |y| > L \\ \text{ExtractAnswer}(x, y) & \text{otherwise} \end{cases}$$

where $y_{:L}$ denotes the first $L$ tokens of the reasoning trace and $\text{ExtractAnswer}(x, y)$ extracts the final answer from a complete trace. This approach allows consistent evaluation across different budgets, though it introduces a minor additional token cost in the truncation case.

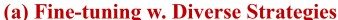

**(a) Fine-tuning w. Diverse Strategies**      **(b) Reasoning Efficiency Frontier**

Figure 1: **Procedure of estimating the reasoning efficiency frontiers.** (a) Starting from a base LRM $\pi_{\theta_{\text{base}}}$, we apply diverse fine-tuning strategies to obtain a large amount of LRMs. (b) We then compute the best achievable accuracy across varying token budgets to obtain the reasoning efficiency frontiers (Eq. 4)

**Length-Constrained Reward and Optimality.**    We define the *length-constrained reward* for a model $\pi_\theta$ over a question distribution $\mathcal{D}$ as the expected reward obtained when the model is restricted to a token budget $L$,

$$J(\mathcal{D}, \theta, L) = \mathbb{E}_{x \sim \mathcal{D}}[\mathbb{E}_{y \sim \pi_\theta(\cdot|x)}[\mathcal{R}(x, \text{Answer}(\pi_\theta, x, y_{:L}))]] \tag{2}$$

The *length-constrained optimal reward* then captures the best possible reward achievable by any model in a parameter space $\Theta$ under the same budget,

$$J_{\text{optimal}}(\mathcal{D}, \Theta, L) = \max_{\theta \in \Theta} J(\mathcal{D}, \theta, L) \tag{3}$$

## 4.2 Empirical Estimation of Reasoning Frontiers

We aim to characterize the optimal reasoning efficiency, denoted by $J_{\text{optimal}}(\mathcal{D}, \Theta, L)$, that reflects the best achievable reward at any token budget $L$ across all possible model parameters $\theta \in \Theta$. However, computing this optimal frontier exactly is infeasible in practice, as it requires exhaustively exploring all algorithms and training configurations. Instead, we construct an empirical reasoning efficiency frontier by fine-tuning a diverse set of models using existing approaches. Let $\hat{\Theta} = \{\theta_1, \ldots, \theta_m\} \subseteq \Theta$ denote the collection of parameters from $m$ fine-tuned models. Based on these, we define the empirical reasoning efficiency frontier as,

> **Definition 4.1: Reasoning Efficiency Frontier**
>
> Given a parameter space $\Theta$, a set of model parameters $\hat{\Theta} = \{\theta_1, \cdots, \theta_m\}$ and a question distribution $\mathcal{D}$, we define the reasoning efficiency frontier as a set of points $\{\hat{J}_{\text{optimal}}(\mathcal{D}, \hat{\Theta}, L) | L \in [1, L_{\max}]\}$ where
>
> $$\hat{J}_{\text{optimal}}(\mathcal{D}, \hat{\Theta}, L) = \max_{\theta \in \{\theta_1, \cdots, \theta_m\}} J(\mathcal{D}_t, \theta, L) \quad \forall L \in [1, L_{\max}] \tag{4}$$

Note that $\hat{J}_{\text{optimal}}(\mathcal{D}, \hat{\Theta}, L)$ serves as a lower bound of the optimal frontier since $\hat{\Theta}$ is a subset of $\Theta$,

$$\hat{J}_{\text{optimal}}(\mathcal{D}, \hat{\Theta}, L) \leq J_{\text{optimal}}(\mathcal{D}, \Theta, L)$$

**Diverse Fine-tuning Approaches.**    To obtain a close approximation to the optimal frontier in Eq. 4, we fine-tune models using a wide range of training strategies,

- **Online RL with Token Budgets:** We conduct online RL training with different token budgets ranging from 512 to 32k. Fine-tuning an LRM with a token budget effectively enforces the LRM to reason with limited tokens [Xu et al., 2025, Hou et al., 2025].

- **Online RL with Length Rewards.** We test various reward designs that promote concise yet accurate reasoning, including length-harmonizing rewards [Luo et al., 2025a] and length-group normalized rewards [Arora and Zanette, 2025].

- **Preference Learning.** We apply SimPO [Meng et al., 2024] with preference datasets constructed via methods such as TOPS [Yang et al., 2025b] and DAST [Shen et al., 2025b]. For example, we contrast short correct responses with longer ones to promote conciseness [Munkhbat et al., 2025].

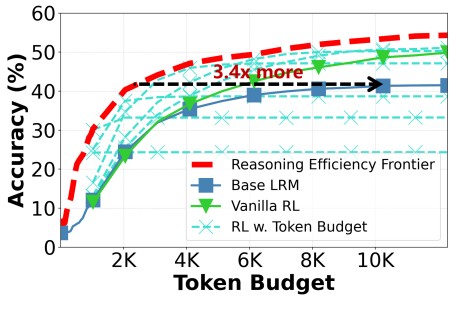 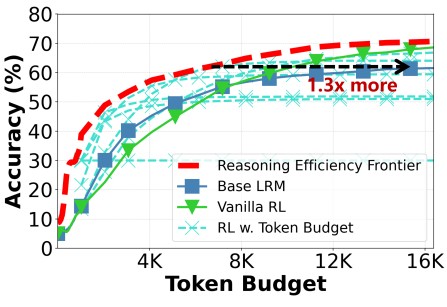

(a) DeepSeek-R1-Distill-Qwen-1.5B  (b) DeepSeek-R1-Distill-Qwen-7B

Figure 2: Reasoning Efficiency Frontiers for DeepSeek-R1-Distill-Qwen1.5B and DeepSeek-R1-Distill-Qwen-7B. DeepSeek-R1-Distill-Qwen-1.5B and DeepSeek-R1-Distill-Qwen-7B spend $3.4\times$ and $1.3\times$ more tokens than the corresponding frontiers to achieve the same accuracies.

**Experiment Setup.** We conduct our study using two base LRMs, *DeepSeek-R1-Distill-Qwen-1.5B* and *DeepSeek-R1-Distill-Qwen-7B* [Guo et al., 2025]. For each LRM, we start from RL-fine-tuned versions and further fine-tune them using the strategies above. We evaluate on three challenging mathematical reasoning benchmarks: AMC 2023, AIME 2024, and AIME 2025. More training details can be found in Sec. 6.1 and Appendix. C. We conducted large-scale RL experiments spanning 8 algorithms and 15 training configurations, resulting in 180+ and 210+ models for 1.5B and 7B scales, respectively.

**Reasoning Efficiency Frontiers.** Based on the fine-tuned models, we are able to construct the reasoning efficiency frontiers for both base LRMs. Notably, we find that DeepSeek-R1-Distill-Qwen-1.5B and DeepSeek-R1-Distill-Qwen-7B require approximately $3.4\times$ and $1.3\times$ more tokens, respectively, than their corresponding empirical frontiers to achieve the same level of accuracy.

**Measuring the Gap to Optimality.** Given the estimated reasoning efficiency frontiers $\hat{J}_{\text{optimal}}(\mathcal{D}, \Theta, L)$, we can use the metric of ***Reasoning Efficiency Gap (REG)*** to quantify the distance of any LRM from reaching the optimal reasoning efficiency.

---

**Definition 4.2: Reasoning Efficiency Gap (REG)**

Given any LRM $\pi_\theta$ and the estimated reasoning efficiency frontier $\{\hat{J}_{\text{optimal}}(\mathcal{D}, \hat{\Theta}, L)|L \in [1, L_{\max}]\}$, we define Reasoning Efficiency Gap as,

$$d_{\text{REG}}(\theta, \mathcal{D}, \hat{\Theta}) = \sum_{L=1}^{L_{\max}} \hat{J}_{\text{optimal}}(\mathcal{D}, \hat{\Theta}, L) - J(\mathcal{D}, \theta, L) \qquad (5)$$

---

## 5 Methodology

### 5.1 Boosting Reasoning Efficiency by Optimizing Length-Constrained Rewards

**Optimizing Length-Constrained Rewards.** To minimize the reasoning efficiency gap, a straightforward idea is to optimize the length-constrained rewards under all token budgets to enhance the reasoning efficiency of $\pi_{\theta_{\text{base}}}$, leading to the efficiency objective,

$$\mathcal{L}_{\text{Efficiency}}(\theta, \mathcal{D}) = \sum_{L=1}^{L_{\max}} J(\mathcal{D}, \theta, L) \qquad (6)$$

where $L_{\max}$ is the maximum generation length. However, directly optimizing Eq. 6 is computationally impractical. Evaluating $J(\mathcal{D}, \theta, L)$ for each budget $L \in [1, L_{\max}]$ requires separate inference runs to evaluate truncated responses as discussed in Sec. 4.1. As a result, each training example would require up to $L_{\max}$ additional LRM generations, significantly increasing both compute time and memory usage, particularly due to the expanded KV cache usage.

## 5.2 Reasoning Efficiency Optimization with Reinforcement Learning

**REO-RL.** We introduce *Reasoning Efficiency Optimization with Reinforcement Learning (REO-RL)*, an efficient training algorithm that optimizes an approximation for the objective in Eq. 6. In REO-RL, instead of optimizing the length-constrained reward in all token budgets, we approximate the objective with a small set of selected token budgets $L_1, \cdots, L_N$ to ensure high training efficiency. Specifically, following the Trapezoidal rule in numerical integration, we could approximate the objective in Eq. 6 with,

$$\sum_{L=1}^{L_{\max}} J(\mathcal{D}, \theta, L) \approx \sum_{i=1}^{N} \frac{L_{i+1} - L_{i-1}}{2} J(\mathcal{D}, \theta, L_i) + \frac{L_1}{2} \cdot J(\mathcal{D}, \theta, 0) + \frac{L_{\max} - L_N}{2} \cdot J(\mathcal{D}, \theta, L_{\max}) \tag{7}$$

$$= f(\mathcal{D}, \theta, \{L_1, \cdots, L_N\}) \tag{8}$$

where we assume $L_0 = 0$ and $L_{N+1} = L_{\max}$ and $f(\mathcal{D}, \theta, \{L_1, \cdots, L_N\})$ denotes the approximated objective with token budgets $L_1, \cdots, L_N$. As the number of selected token budgets $N$ increases, $f(\mathcal{D}, \theta, \{L_1, \cdots, L_N\})$ would become closer and closer to the objective in Eq. 6.

The approximated objective $f(\mathcal{D}, \theta, \{L_1, \cdots, L_N\})$ could be equivalently represented as an RL objective with dense rewards,

$$\mathcal{L}_{\text{approx}}(\theta, \mathcal{D}) = \mathbb{E}_{x \sim \mathcal{D}} \left[ \mathbb{E}_{y \sim \pi_\theta(\cdot|x)} \left[ \sum_{i=1}^{N+1} c_i \mathcal{R}(x, \text{Answer}(\pi_\theta, x, y_{:L_i})) \right] \right] \tag{9}$$

where $c_i = \frac{L_{i+1} - L_{i-1}}{2}$ for $1 \leq i \leq N$ and $c_{N+1} = \frac{L_{\max} - L_N}{2}$ are the coefficient for the $i$-th token budget following Eq. 8.

Directly computing the derivative of $\theta$ in Eq. 9 would lead to the following equation with two terms,

$$\nabla_\theta \mathcal{L}_{\text{approx}}(\theta, \mathcal{D}) = \mathbb{E}_{x \sim \mathcal{D}} \left[ \underbrace{\nabla_\theta \mathbb{E}_{y \sim \pi_\theta(\cdot|x)} \left[ \sum_{i=1}^{N+1} c_i \cdot \mathcal{R}(x, \text{Answer}(\pi_{\theta'}, x, y_{:L_i})) \right]}_{\text{Rewards of Truncated Responses}} \right] \tag{10}$$

$$+ \mathbb{E}_{x \sim \mathcal{D}} \left[ \underbrace{\mathbb{E}_{y \sim \pi_{\theta'}(\cdot|x)} \left[ \sum_{i=1}^{N+1} c_i \cdot \nabla_\theta \mathcal{R}(x, \text{Answer}(\pi_\theta, x, y_{:L_i})) \right]}_{\text{Probability of Outputting Correct Answer Directly Given Truncated Responses}} \right] \tag{11}$$

where $\theta' = \text{sg}(\theta)$ is equivalent to $\theta$ but does not propagate gradients. Note that the first term can be computed through standard reinforcement learning algorithm. On the other hand, the second term focuses on direct answer generation from truncated responses. In practice, we neglect the answer generation term since this term focuses on a much smaller amount of tokens compared with the first term and we also find optimizing answer generation not bringing empirical benefits.

**REO-RL.** To this end, we propose REO-RL that focuses on improving the rewards of responses truncated under different token budgets. Suppose we use REINFORCE algorithm, the objective of REO-RL is derived by

$$\nabla_\theta \mathcal{L}_{REO-RL}(\theta, \mathcal{D}) =$$

$$\mathbb{E}_{x \sim \mathcal{D}, y \sim \pi_{\theta'}(\cdot|x)} \left[ \sum_{i=1}^{N+1} \nabla_\theta \pi_\theta(y_{L_{i-1}:L_i}|x, y_{:L_{i-1}}) \cdot \sum_{j=i}^{N+1} c_j \mathcal{R}(x, \text{Answer}(\pi_{\theta'}, x, y_{:L_j})) \right]$$

In Eq. 8, different sets of token budgets $L_1, \cdots, L_N$ would cause different approximation error . Ideally, the approximation error induced by the selected token budgets should be low to ensure that REO-RL aligns with the original objective in Eq. 4.

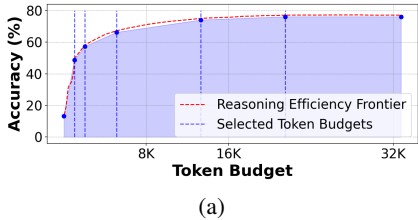 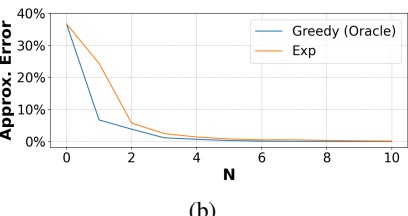

(a)                                        (b)

Figure 3: Selection of Token Budgets in REO-RL. (a) Token budgets selected roughly follow an exponential pattern. (b) Both the oracle greedy approach and the exponential approach achieve lower approximation errors with a few token budgets.

**REO-RL (Oracle).** Note that the original objective in Eq. 4 is bounded by the theoretical reasoning efficiency frontier in Sec. 4.2, i.e. $\sum_{L=1}^{L_{\max}} J(\mathcal{D}, \theta, L) \leq \sum_{L=1}^{L_{\max}} J_{\text{optimal}}(\mathcal{D}, \Theta, L)$. A natural idea is to determine the optimal token budget selection scheme based on the estimated reasoning efficiency frontiers. Specifically, for any set of token budgets $L_1, L_2, \cdots, L_N$, $\sum_{L=1}^{L_{\max}} \hat{J}_{\text{optimal}}(\mathcal{D}, \hat{\Theta}, L)$ could be approximated in a similar way as Eq. 8,

$$f_{\text{optimal}}(\mathcal{D}, \hat{\Theta}, \{L_1, \cdots, L_N\}) = \sum_{i=1}^{N} \frac{L_{i+1} - L_{i-1}}{2} \hat{J}_{\text{optimal}}(\mathcal{D}, \hat{\Theta}, L_i) + \frac{L_1}{2} \cdot \hat{J}_{\text{optimal}}(\mathcal{D}, \hat{\Theta}, 0) \quad (12)$$

$$+ \frac{L_{\max} - L_N}{2} \cdot \hat{J}_{\text{optimal}}(\mathcal{D}, \hat{\Theta}, L_{\max}) \quad (13)$$

where $f_{\text{optimal}}(\mathcal{D}, \hat{\Theta}, \{L_1, \cdots, L_N\})$ denotes the approximated value for $\sum_{L=1}^{L_{\max}} \hat{J}_{\text{optimal}}(\mathcal{D}, \hat{\Theta}, L)$ given token budgets $L_1, \cdots, L_N$.

We adopt a greedy approach that iteratively selects the token budget with the lowest approximation error. Note that this is an oracle approach since the reasoning efficiency frontiers should be known in advance. This greedy selection approach leads to an oracle algorithm, **_REO-RL (Oracle)_**. In REO-RL (Oracle), the token budgets $L_1, \cdots, L_N$ are selected according to,

$$L_i = \arg\min_{L'} \mid f_{\text{optimal}}(\mathcal{D}, \hat{\Theta}, \{L_1, \cdots, L_{i-1}, L'\}) - \sum_{L=1}^{L_{\max}} \hat{J}_{\text{optimal}}(\mathcal{D}, \hat{\Theta}, L) \mid$$

The oracle greedy approach could produce a set of token budgets that achieves low approximation error in Fig. 3(a). As illustrated in Fig. 3(b), the approximation error gradually degrades with more token budgets. Notably, the approximation error could be lower than $1\%$ with $N \geq 5$.

**REO-RL (Exp).** However, in cases when the reasoning efficiency frontiers are unknown, it is infeasible to apply REO-RL (Oracle). We observe that token budgets selected by the greedy selection approach roughly follow an exponentially spaced pattern as shown in Fig. 3(a). Therefore, we propose to adopt an exponentially spaced scheme for token budget selection, leading to the algorithm, **_REO-RL (Exp)_**, that selects a set of exponentially spaced token budgets,

$$L_i = L_{\min} \cdot (L_{\max}/L_{\min})^{\frac{i-1}{N}}$$

where $L_{\min}/L_{\max}$ are the minimum/maximum token budgets.

## 6    Experiments

### 6.1    Experimental Setup

**Models, Datasets & Metrics.** We use DeepSeek-R1-Distill-Qwen-1.5B and DeepSeek-R1-Distill-Qwen-7B as the base LRMs. For training, we adopt a mixture of training data consisting of 135k problems sourced from DeepScaleR Luo et al. [2025c] and AReaL RL Lab [2025] For evaluation, we use three challenging mathematical benchmarks: AMC 2023, AIME 2024, and AIME 2025. We report the average accuracy of 32 responses generated with temperature $T = 0.6$ and top_p$= 0.95$ with maximum length $L_{\max} = 32K$. The main results are averaged over three benchmarks. When evaluating REG, we use a smaller length $L_{\max} = 16K$ to focus on the area under lower token budgets.

**REO-RL & Baselines.** For REO-RL, we consider both REO-RL (Exp), REO-RL (Oracle), and a variant of REO-RL (Exp) that uniformly sets all coefficients $c_i = 1$. For baselines, we consider,

- **Online RL**: We consider online RL with length-based rewards, including length-harmonizing rewards [Luo et al., 2025a], and length group normalized rewards [Arora and Zanette, 2025]. We also compare with Meta Reinforcement Fine-tuning (MFT) [Qu et al., 2025], which minimizes regret for individual steps to enhance reasoning efficiency. Additionally, we also adopt online RL with hard token budgets as a baseline, which is also adopted in recent works [Hou et al., 2025, Xu et al., 2025].

- **Supervised Fine-Tuning**: For each problem in the training data, we generate multiple responses and perform SFT on the shortest correct one. Two strategies are used for data generation. The first strategy is direct generation with the problems as inputs [Munkhbat et al., 2025]. The second strategy follows TOPS to prompt the LRM with different levels of reasoning efforts [Yang et al., 2025b].

- **Preference Learning**: We apply SimPO [Meng et al., 2024] on various preference datasets. In $SimPO_{shortest}$, we use the shortest correct response and the longest response as the preference pairs. We also follow TOPS [Yang et al., 2025b] and DAST [Shen et al., 2025b] to construct preference datasets.

We have also tried RL-based length control methods such as [Aggarwal and Welleck, 2025, Xu et al., 2025] but find these methods only achieve successful length control under low token budgets and yield similar performance as RL w. Token Budgets. More details about the baselines and our other investigations could be found in Appendix. D.

**Training Details.** For REO-RL and all baselines, we use the same training data and implement all methods based on the AReaL framework. Instead of directly fine-tuning the base LRMs, we fine-tune the corresponding RL-trained versions. Specifically, we use AReaL-Boba-RL-1.5B [RL Lab, 2025] and SkyWork-OR1-Math-7B [He et al., 2025] as the starting points for further fine-tuning in 1.5B and 7B experiments, respectively. For REO-RL (Exp) and REO-RL (Oracle), we use $N = 5$ token budgets since $N = 5$ already obtains sufficiently accurate approximations as illustrated in Fig. 3(b). For more training details, please refer to Appendix. C.

## 6.2 Main Results

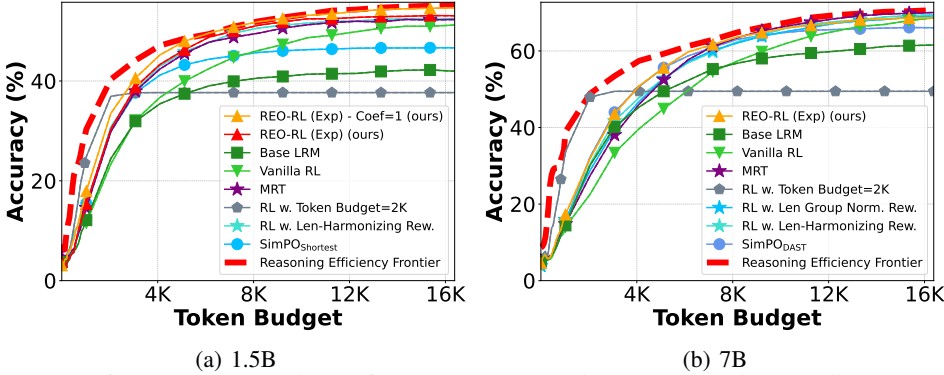

(a) 1.5B          (b) 7B

Figure 4: Performance comparison of REO-RL (Exp) with representative baseline methods on DeepSeek-R1-Distill-Qwen-1.5B and DeepSeek-R1-Distill-Qwen-7B. REO-RL (Exp) notably improves the reasoning efficiency of LRMs. Although existing approaches can approach the efficiency frontier when sufficient token budgets are available, there still exist large performance gaps under tight token constraints. Results are averaged over AMC 2023, AIME 2024 and AIME 2025.

Fig. 4 presents the accuracy of LRMs fine-tuned using REO-RL (Exp) and a range of strong baseline methods across varying token budgets.[2] Full quantitative results are detailed in Tab. 1. We draw several key observations from the experiments,

**There exist fundamental gaps between existing approaches and the frontiers (Fig. 4).** As shown in Fig. 4, several existing approaches can approach or even reach the reasoning efficiency frontier

---

[2]For clearer visualization, we include only a set of representative approaches and plot their performance under limited token budgets.

| Method | DeepSeek-R1-Distill-Qwen-1.5B | | | DeepSeek-R1-Distill-Qwen-7B | | |
|---|---|---|---|---|---|---|
| | Accuracy (%) ↑ | Length ↓ | REG ↓ | Accuracy (%) ↑ | Length ↓ | REG ↓ |
| Base LRM | 41.4 | 14430.5 | 2036.7 | 62.0 | 11160.4 | 1782.1 |
| Vanilla RL | 51.6 | 11635.5 | 1257.0 | 71.0 | 12609.9 | 1691.6 |
| REO-RL (Exp) (ours) | 53.1$_{↑1.5}$ | 7467.2$_{↓35.8\%}$ | **551.8**$_{↓56.1\%}$ | 68.7$_{↓2.3}$ | 6725.7$_{↓46.7\%}$ | 816.7$_{↓51.7\%}$ |
| REO-RL (Exp) - Coef=1 (ours) | 54.3$_{↑2.7}$ | 9042.7$_{↓22.3\%}$ | **320.8**$_{↓74.5\%}$ | 67.6$_{↓3.4}$ | 6160.3$_{↓51.1\%}$ | 605.9$_{↓64.2\%}$ |
| REO-RL (Oracle) (ours) | 53.9$_{↑2.2}$ | 7119.7$_{↓38.8\%}$ | **606.9**$_{↓51.7\%}$ | 69.2$_{↓1.9}$ | 7660.8$_{↓39.2\%}$ | 771.1$_{↓54.4\%}$ |
| RL w. Token Budget=1K | 32.2$_{↓19.5}$ | 1293.9$_{↓88.9\%}$ | 2823.6$_{↑124.6\%}$ | 39.8$_{↓31.2}$ | 1123.6$_{↓91.1\%}$ | 3627.1$_{↑114.4\%}$ |
| RL w. Token Budget=2K | 37.7$_{↓13.9}$ | 1979.0$_{↓83.0\%}$ | 2077.5$_{↑65.3\%}$ | 49.5$_{↓21.6}$ | 1729.6$_{↓86.3\%}$ | 2256.5$_{↑33.4\%}$ |
| RL w. Token Budget=4K | 46.1$_{↓5.5}$ | 3441.0$_{↓70.4\%}$ | 1045.3$_{↓16.8\%}$ | 57.6$_{↓13.4}$ | 2978.4$_{↓76.4\%}$ | 1343.3$_{↓20.6\%}$ |
| RL w. Len Group Norm. Rew. | 52.6$_{↑1.0}$ | 8999.0$_{↓22.7\%}$ | 891.2$_{↓29.1\%}$ | 69.8$_{↓1.2}$ | 8961.3$_{↓28.9\%}$ | 1057.2$_{↓37.5\%}$ |
| RL w. Len-Harmonizing Rew. | 52.7$_{↑1.1}$ | 7075.4$_{↓39.2\%}$ | 622.8$_{↓50.4\%}$ | 70.4$_{↓0.7}$ | 7956.8$_{↓36.9\%}$ | 970.8$_{↓42.6\%}$ |
| MRT | 53.4$_{↑1.8}$ | 9031.8$_{↓22.4\%}$ | 770.2$_{↓38.7\%}$ | 70.1$_{↓0.9}$ | 8252.2$_{↓34.6\%}$ | 976.4$_{↓42.3\%}$ |
| SFT$_{Shortest}$ | 51.9$_{↑0.3}$ | 11544.5$_{↓0.8\%}$ | 1166.0$_{↓7.2\%}$ | 71.6$_{↑0.6}$ | 11711.3$_{↓7.1\%}$ | 1542.4$_{↓8.8\%}$ |
| SFT$_{TOPS}$ | 51.7$_{↑0.1}$ | 9666.2$_{↓16.9\%}$ | 1090.6$_{↓13.2\%}$ | 70.4$_{↓0.6}$ | 11549.1$_{↓8.4\%}$ | 1519.3$_{↓10.2\%}$ |
| SimPO$_{DAST}$ | 36.8$_{↓14.8}$ | 10382.2$_{↓10.8\%}$ | 2318.3$_{↑84.4\%}$ | 66.1$_{↓4.9}$ | 6489.7$_{↓48.5\%}$ | 987.8$_{↓41.6\%}$ |
| SimPO$_{Shortest}$ | 46.6$_{↓5.0}$ | 6274.6$_{↓46.1\%}$ | 1250.7$_{↓0.5\%}$ | 69.9$_{↓1.1}$ | 8144.5$_{↓35.4\%}$ | 1060.3$_{↓37.3\%}$ |
| SimPO$_{TOPS}$ | 30.8$_{↓20.8}$ | 2185.8$_{↓81.2\%}$ | 3137.0$_{↑149.6\%}$ | 58.3$_{↓12.7}$ | 4379.3$_{↓65.3\%}$ | 1453.3$_{↓14.1\%}$ |

Table 1: Accuracy, Generation Length, and Reasoning Efficiency Gap (REG) for all methods. The relative changes compared to the vanilla RL baseline are also reported. REO-RL could significantly reduce the gap from the LRM to the reasoning efficiency frontier with minor or even no accuracy drop at the same time.

given sufficient token budgets. However, they could require significantly more tokens than the efficiency frontier to achieve moderate-level accuracies. For example, on On the other hand, some methods, such as RL with Token Budget, perform well under tight token constraints but exhibit substantially lower overall accuracy. Our benchmarking results suggest that, *optimizing the base LRM to precisely match reasoning efficiency frontiers remains an open problem.*

**REO-RL consistently outperforms baselines in reasoning efficiency (Tab. 1).** Compared to baseline methods, the three variants of REO-RL are much closer to the reasoning efficiency frontier, exhibiting much lower Reasoning Efficiency Gap (REG). REO-RL maintains high accuracy while generating shorter outputs. REO-RL (Exp) - Coef=1 performs the best in terms of reasoning efficiency, reducing the REG by 74.5% and 64.2% for the 1.5B and 7B models, respectively.

**Vanilla RL does not yield consistent improvements across budgets (Fig. 4).** Although Vanilla RL improves overall accuracy relative to the base LRM, it fails to provide stable gains across different token budgets. In the 7B experiments, its accuracy is lower than the base LRM when the token budget is below 8K.

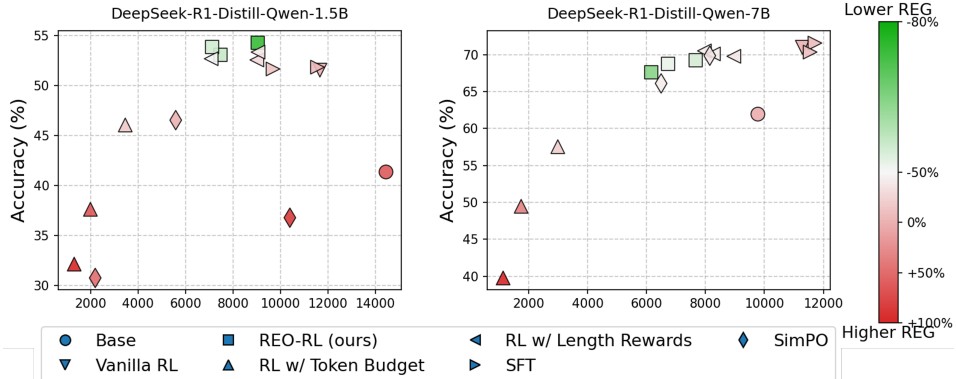

Figure 5: **REG effectively captures the trade-off between accuracy and response length.** Achieving a low REG require both competitive accuracy and short response length. By minimizing the efficiency gap, REO-RL outperforms baselines in terms of reasoning efficiency.

**REG effectively captures the trade-off between accuracy and response length (Fig. 5).** Achieving low REG requires balancing high accuracy with concise output. For example, REO-RL (Exp) maintains strong accuracy while generating compact responses, resulting in low REG. In contrast, methods that optimize only one aspect, such as minimizing length at the cost of accuracy, fail to achieve low REG. RL with Token Budgets, despite producing shorter responses, suffers from significant accuracy loss.

## 6.3 Comparison with Frontier LRMs

To further evaluate the reasoning efficiency of LRMs trained with REO-RL (Exp) in comparison to advanced LRMs, we conduct a controlled analysis focused on the correct response length. Since frontier LRMs and those fine-tuned with REO-RL (Exp) differ significantly in overall accuracy, we construct a balanced subset of 71 questions from the test set, where all models achieve an accuracy exceeding 50%. For each model, we compute the average length of correct reasoning traces for each problem. The final length metric for each model is then obtained by averaging the correct response lengths across all 71 questions. As shown in Tab. 2, the 7B model trained with REO-RL (Exp) exhibits more concise reasoning patterns than frontier LRMs, with a much shorter response length.

| | Claude Sonnet 3.7 (Thinking) | DeepSeek R1 | Qwen3-4B | Qwen3-8B | Qwen3-32B | Vanilla RL - 7B | REO-RL (Exp) - 7B (ours) |
|---|---|---|---|---|---|---|---|
| Length | 17478.87 | 5156.39 | 8631.61 | 8898.57 | 7755.99 | 7732.11 | **4524.02** |
| Accuracy | 90.0% | 98.2% | 95.8% | 94.9% | 97.2% | 93.5% | 93.5% |

Table 2: Comparing the response length with frontier LRMs.

## 6.4 Ablation Study of REO-RL

We conduct an ablation study on the design choices of REO-RL using the DeepSeek-R1-Distill-Qwen-7B model to better understand its flexibility and performance characteristics. By varying key components, we demonstrate that REO-RL can maintain competitive performance across different configurations.

**Token budget selection strategy.** We evaluate REO-RL (Oracle) that adopts the oracle greedy strategy. This variant achieves slightly better performance than REO-RL (Exp), reflected in a lower REG. We also investigate linearly spaced token budgets, which leads to less optimal efficiency and higher REG.

| Method | Accuracy (%) ↑ | Length ↓ | REG |
|---|---|---|---|
| Vanilla RL | 71.0 | 12609.9 | 1691.6 |
| Base LRM | 62.0 | 11160.4 | 1782.1 |
| Exp | $68.7_{\downarrow 2.3}$ | $6725.7_{\downarrow 46.7\%}$ | $816.7_{\downarrow 51.7\%}$ |
| Oracle | $69.2_{\downarrow 1.9}$ | $7660.8_{\downarrow 39.2\%}$ | $771.1_{\downarrow 54.4\%}$ |
| Exp - Coef=1 | $67.6_{\downarrow 3.4}$ | $6160.3_{\downarrow 51.1\%}$ | $605.9_{\downarrow 64.2\%}$ |
| Exp - N=10 | $69.0_{\downarrow 2.0}$ | $7979.1_{\downarrow 36.7\%}$ | $923.2_{\downarrow 45.4\%}$ |
| Linear | $69.1_{\downarrow 1.9}$ | $7424.8_{\downarrow 41.1\%}$ | $886.1_{\downarrow 47.6\%}$ |
| Question-Specific Oracle | $69.5_{\downarrow 1.5}$ | $7312.3_{\downarrow 42.0\%}$ | $687.7_{\downarrow 59.3\%}$ |

Table 3: Ablation Study of REO-RL on DeepSeek-R1-Distill-Qwen-7B.

**Coefficient $c_i$ in REO-RL objective.** Setting all coefficients $c_i$ uniformly to 1 leads the model to align more closely with the efficiency frontier and shorter response lengths. However, this approach comes at the cost of reduced overall accuracy.

**Number of selected token budgets $N$.** We increase the number of selected token budgets to $N = 10$ to explore whether a finer granularity improves performance. In practice, we observe that using more token budgets results in slower convergence and a higher rate of training instability. Consequently, this configuration produces weaker results overall.

**Question-Specific Oracle Budgets.** Finally, we investigate a question-specific oracle strategy that assigns two token budgets to each problem, the minimum reasoning length derived through the frontier estimation experiments (Sec. 4.2) and the full budget $L_{\max}$. This oracle approach proves to be competitive, further reducing the efficiency gap of REO-RL.

## 7 Conclusion

In this work, we investigate efficient reasoning for LRMs. We introduce reasoning efficiency frontiers, which characterize the empirically optimal trade-off between response length and accuracy for LRMs. To quantify the reasoning efficiency of a fine-tuned LRM, we introduce the Reasoning Efficiency Gap (REG), a unified metric that captures both accuracy and length. We benchmark existing methods and reveal a substantial gap between current fine-tuning approaches and the frontiers. Our proposed method, REO-RL, consistently obtains better reasoning efficiency than strong baselines across model scales. Despite these gains, achieving full match with the reasoning efficiency frontiers remains an open problem.

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

## A  Limitations

**Limitations.**  Our study is currently only taken place on 1.5B and 7B LRMs. Extending our research to larger and more powerful models could potentially uncover more efficient reasoning capabilities. Additionally, our evaluations and training are primarily based on mathematical reasoning tasks. Broadening our scope to include more diverse tasks, such as competitive programming and multimodal reasoning, may enhance the generalizability and impact of our approach.

## B  Reproducibility

We provide our code in the https://anonymous.4open.science/r/REO-RL-803F. Please refer to Sec. F for the details on reasoning efficiency frontiers and reasoning efficiency gap, and Sec. C for the implementation detials.

## C  Implementation Details

**Training Data.**  For training data, we integrate data from DeepScaleR Luo et al. [2025c] and AReaL RL Lab [2025]. For 7B we directly use the training data of AReaL-Boba-RL-7B [RL Lab, 2025]. For 1.5B, we adopt the mixture of training data from DeepScaleR Luo et al. [2025c] and AReaL RL Lab [2025] and remove duplicated problems.

We implement the training algorithm with the AReaL framework [RL Lab, 2025], which supports SGLang [?] for rollout generation. Below we detail implementation details of each training algorithm.

**Online RL Training.**  We use PPO as the default online RL algorithm. Following standard practices in RL training for LLM reasoning [Yu et al., 2025b, Hu et al., 2025], we do not utilize value model and KL regularization. The default training setting and hyperparameters for PPO training are listed in Tab. 4.

For REO-RL and baseline methods, we do not carry out RL training directly from the base LRM since it would lead to prolonged training and slower convergence. Instead, we perform further fine-tuning on the RL trained versions for both 1.5B and 7B settings. Specifically, we adopt AReaL-Boba-RL-1.5B [RL Lab, 2025] and Skywork-OR1-Math-7B [He et al., 2025] as the starting points for further fine-tuning. Following the cluster config in Tab. 4, each experiment could finish within 48 hours.

**Supervised Fine-Tuning.**  The default training configurations and hyperparameters for SFT are listed in Tab. 5.

**Preference Learning.**  We implement SimPO [Meng et al., 2024] in the AReaL framework [RL Lab, 2025]. The default training configurations and hyperparameters for SimPO are listed in Tab. 5.

## D  Baselines

**RL with Token Budgets.**  In our online RL training with token budget constraints, we control the maximum generation length during each training phase. Rather than fine-tuning the LRM directly on a fixed token budget, we adopt a progressive length-shrinking strategy. We begin training with a 16K token budget. Once RL training at this level converges, we reduce the budget to 8K and continue training. This process is repeated, halving the token budget each time, until we reach the minimum budget of 512 tokens. This staged approach enables the model to gradually adapt to shorter generation lengths while maintaining reasoning performance.

**RL with Length Rewards.**  In the "RL with Length Group Normalized Rewards" baseline [Arora and Zanette, 2025], for each question $x$ and the corresponding set of sampled responses $y^1, \cdots, y^m$, the reward of response $y^i$ is computed as,

$$r(x, y^i) = \mathbb{I}\{y^i \text{ is correct}\}(1 - \alpha f(|y^i|))$$

Table 4: Default training configurations and hyperparameters for PPO.

| Training Configuration | |
| --- | --- |
| Batch size (number of prompts) | 512 |
| Rollouts per prompt | 16 |
| Random seed | 1 |
| Cluster Config | $8 \times 8$ H800 (for 1.5B) / $16 \times 8$ H800 (for 7B) |
| **PPO Parameters** | |
| PPO Minibatches | 4 |
| Clipping $\epsilon$ | 0.2 |
| Advantage normalization | True |
| Discount factor $\gamma$ | 1.0 |
| GAE $\lambda$ | 1.0 |
| Epochs | 2.0 |
| **Optimizer Parameters** | |
| Optimizer | Adam |
| Learning rate | $2.0 \times 10^{-5}$ |
| Weight decay | 0.05 |
| $\beta_1$ | 0.9 |
| $\beta_2$ | 0.95 |
| Adam $\epsilon$ | $1 \times 10^{-5}$ |
| Gradient norm clipping | 1.0 |
| Learning rate scheduler | constant |
| Warmup steps proportion | 0.001 |
| **Generation Parameters** | |
| Temperature | 1.0 |
| Top-p | 1.0 |
| Top-k | -1 |
| Max prompt length | 1024 |
| Min generation length | 0 |
| Max generation length | 24376 (for 1.5B) / 32768 (for 7B) |

where the function $f$ normalizes $|y^i|$ according to the lengths of correct responses and applies a sigmoid function. Specifically,

$$f(|y^i|) = \sigma \left( \frac{|y^i| - \texttt{MEAN}(x)}{\texttt{STD}(x)} \right)$$

where

$$\texttt{MEAN}(x) = \mathbb{E}_{y \sim \pi(\cdot|x), s.t. y \text{ is correct}}[|y|]$$

$$\texttt{STD}(x) = \sqrt{\text{Var}_{y \sim \pi(\cdot|x), s.t. y \text{ is correct}}[|y|]}$$

In the "RL with Length-Harmonizing Rewards" baseline [Luo et al., 2025a], for each question $x$ and the corresponding set of sampled responses $y^1, \cdots, y^m$, the reward of response $y^i$ is computed as,

$$r(x, y^i) = \frac{\overline{L}_{ref}(x)}{|y|} - 1 + \gamma \cdot (\mathbb{I}\{y \text{ is correct}\} - \overline{A}_{ref}(x))$$

where $\overline{L}_{ref}(x)$ is the average response length of the reference model when taking $x$ as input and $\overline{A}_{ref}(x)$ is the average accuracy of the reference model. In our implementations, we set the models that serve as the starting point of RL training as the reference models.

In the MRT baseline [Qu et al., 2025], different from the original paper that only implements single-step optimization with offline collected response prefixes, we implement the online RL training version with dense rewards for MRT. For each question $x$ and the corresponding set of sampled

Table 5: Default training configurations and hyperparameters for SFT.

| Training Configuration | |
| --- | --- |
| Batch size (number of prompt-answer pairs) | 512 |
| Cluster Config | $16 \times 8$ H800 |
| **SFT Parameters** | |
| Epochs | 10 |
| Save Frequency Steps | 100 |
| use_bf16 | True |
| Max Seq Length | 32768 |
| **Optimizer Parameters** | |
| Optimizer | Adam |
| Learning rate | $1 \times 10^{-5}$ |
| Weight decay | 0.05 |
| $\beta_1$ | 0.9 |
| $\beta_2$ | 0.95 |
| Adam $\epsilon$ | $1 \times 10^{-5}$ |
| Gradient norm clipping | 1.0 |
| Learning rate scheduler | constant |
| Warmup steps proportion | 0.03 |

Table 6: Default training configurations and hyperparameters for SimPO.

| Training Configuration | |
| --- | --- |
| Batch size (number of preference pairs) | 128 |
| Cluster Config | $16 \times 8$ H800 |
| **SimPO Parameters** | |
| Epochs | 2 |
| Save Frequency Steps | 10 |
| use_bf16 | True |
| Max Seq Length | 32768 |
| SimPO Coefficient $\beta$ | 1/2 |
| SimPO Coefficient $\gamma$ | 1.2/1.4 |
| **Optimizer Parameters** | |
| Optimizer | Adam |
| Learning rate | $1 \times 10^{-5}$ (for 1.5B) / $3 \times 10^{-6}$ (or 7B) |
| Weight decay | 0.05 |
| $\beta_1$ | 0.9 |
| $\beta_2$ | 0.95 |
| Adam $\epsilon$ | $1 \times 10^{-5}$ |
| Gradient norm clipping | 1.0 |
| Learning rate scheduler | constant |
| Warmup steps proportion | 0.03 |

responses , we partition each response into several steps $y = (y_1, \cdots, y_s)$. In each training step, the model is updated by computing the policy gradient for the following objective,

$$\mathbb{E}_{x,y=(y_1,\cdots,y_s)\sim\pi_{\theta'}(\cdot|x)}\Big[\sum_{i=1}^{s}\mathbb{E}_{y_i'\sim\pi_\theta(\cdot|x,y_{:i-1})}[\mathcal{R}(x, \text{Answer}(\pi_{\theta'}, x, [y_{:i-1}; y']))$$
$$- \mathcal{R}(x, \text{Answer}(\pi_{\theta'}, x, y_{:i-1})) + \alpha \cdot \mathcal{R}(x,y)]\Big]$$

where $\alpha$ is the weight for the overall accuracy and is set as $0.2$ in our experiments.

**Supervised Fine-Tuning.** In SFT$_{\text{Shortest}}$, we generate 16 outputs for each question in the training dataset. Then we select the correct response with the shortest length for each question to construct the SFT dataset. In SFT$_{\text{TOPS}}$, we follow [Yang et al., 2025b] to prompt the LRM to generate responses with three different types of reasoning efforts. We strictly follow the prompts used in [Yang et al., 2025b]. For each type of reasoning effort, we generate 16 responses. To construct the SFT dataset, the shortest correct response among all 48 responses are gathered.

**Preference Learning.** We adopt three strategies for constructing the preference datasets. In SimPO$_{\text{Shortest}}$, we adopt the responses generated for SFT$_{\text{Shortest}}$ and select the shortest correct response and the longest response as the preference pair for each question. In SimPO$_{\text{TOPS}}$, we use the same preference construction strategy as SimPO$_{\text{Shortest}}$ but on the responses generated through TOPS, which contain reasoning traces with different reasoning efforts. Finally, in SimPO$_{\text{DAST}}$, we again utilize the responses generated for SFT$_{\text{Shortest}}$ but adopt a different preference construction strategy. In the preference dataset of SimPO$_{\text{DAST}}$, each pair falls into one of the two cases: it either contains two correct responses where the positive sample is much shorter than the negative sample, or contains two incorrect responses where the positive sample is much longer than the negative sample.

**Other Methods We Have Tried.** We have also experimented with Z1 [Yu et al., 2025c] that constructs code-based reasoning traces, and L1 [Aggarwal and Welleck, 2025] that fine-tunes the LRMs to follow token budget instruction with RL. We find Z1 having poor performance on mathematical reasoning tasks, demonstrating significantly lower accuracy to that of the base LRMs as illustrated in Tab. 7. For L1, we find L1 mainly works under tight token budgets, i.e. less than $4K$. When we extend the training approach of L1 to a larger context length, i.e. 24K for 1.5B models, we find it hard to make the LRM learning to follow strict token budget instructions through RL. Consequently, the resulted models, L1-Exact-24K-1.5B and L1-Max-24K-1.5B could not follow the token budget instruction, as shown in Tab, 8.

| Method | AIME24 | | MATH500 | | GPQA | |
|---|---|---|---|---|---|---|
| | Accuracy (%) | Length | Accuracy (%) | Length | Accuracy (%) | Length |
| Base LRM | 31.5 | 16747.6 | 83.6 | 5633.1 | 44.6 | 10325.2 |
| Z1 | 10.0 | 15106.0 | 63.6 | 4904.1 | 61.6 | 9004.1 |

Table 7: Result of Z1 on DeepSeek-R1-Distill-Qwen-1.5B. [Yu et al., 2025c]

| Instructed Token Budget | AMC23 | | AIME24 | | AIME25 | |
|---|---|---|---|---|---|---|
| | Accuracy (%) | Length | Accuracy (%) | Length | Accuracy (%) | Length |
| 2048 | 63.9 | 20267.7 | 38.6 | 18067.4 | 27.2 | 19657.7 |
| 4096 | 63.5 | 20142.6 | 38.4 | 17961.3 | 26.9 | 19711.1 |
| 8192 | 62.9 | 20191.6 | 38.4 | 17868.7 | 26.8 | 19673.5 |

Table 8: Result of L1-Exact-24K-1.5B.

# E  REO-RL

## E.1  Implementing REO-RL

**Generation Phase.** In the generation phase of REO-RL, there are two rounds of LRM generation. In the first round, multiple responses are generated for each question in the training batch. In the second round, to compute the length-constrained rewards for each of the responses and across all selected token budgets, we choose all truncated responses $y_{:L_i}$ and apply a prompt to enforce the LRM to generate the final answer given incomplete reasoning traces, i.e. $a = \pi_\theta(\cdot|x, y_{:L}, [\text{The Final Answer is}])$. We follow the prompt employed by [Fu et al., 2024] and [Fu et al., 2025].

**Prompt for Forcing LRM to Produce Answer**

...
Oh, I suddenly got the answer to the whole problem. **Final Answer**:
[\boxed{

**Dense Reward RL.** REO-RL obtains dense rewards through forcing the LRM to generate answer under various token budgets. The objective of REO-RL is as follows,

$$\textbf{REO-RL:} \quad \mathscr{L}_{REO-RL}(\theta, \mathcal{D}) = \mathbb{E}_{x \sim \mathcal{D}} \left[ \mathbb{E}_{y \sim \pi_\theta(\cdot|x)} \left[ \sum_{i=1}^{N+1} c_i \mathcal{R}(x, \text{Answer}(\pi_\theta, x, y_{:L_i})) \right] \right]$$

where $c_i = \frac{L_{i+1} - L_{i-1}}{2}$ for $1 \leq i \leq N$ and $c_{N+1} = \frac{L_{\max} - L_N}{2}$ are the coefficient for the $i$-th token budget.

To perform policy update, we compute the return for each section between two consecutive token budgets $L_i$ and $L_{i+1}$. For $1 \leq i \leq N$, we compute,

$$\text{Return}_i(x, y) = \sum_{j=i}^{N+1} c_j \mathcal{R}(x, \text{Answer}(\pi_\theta, x, y_{:L_j}))$$

Since we disable value model in the training process, the computed returns are then used directly as the advantages for PPO loss computation. Specifically $\text{Return}_i(x, y)$ would be used to update the tokens $y_{L_i:L_{i+1}}$.

# F  Reasoning Efficiency Frontiers & Reasoning Efficiency Gap

**Reasoning Efficiency Frontiers.** The two boxes below record the detailed lengths and accuracies for points on the estimated reasoning efficiency frontiers for DeepSeek-R1-Distill-Qwen-1.5B and DeepSeek-R1-Distill-Qwen-7B, respectively.

**Reasoning Efficiency Frontier for DeepSeek-R1-Distill-Qwen-1.5B**

[(0, 0.0722), (64, 0.0616), (128, 0.0626), (192, 0.0928), (256, 0.1121), (320, 0.1261), (384, 0.1562), (448, 0.1884), (512, 0.2126), (576, 0.2219), (640, 0.2296), (704, 0.237), (768, 0.2486), (832, 0.2646), (896, 0.2809), (960, 0.2892), (1024, 0.3034), (2048, 0.4035), (3072, 0.4413), (4096, 0.471), (5120, 0.4851), (6144, 0.4941), (7168, 0.5088), (8192, 0.5195), (9216, 0.5273), (10240, 0.5339), (11264, 0.541), (12288, 0.5431), (13312, 0.5469), (14336, 0.5491), (15360, 0.5506), (16384, 0.5517), (17408, 0.552), (18432, 0.5526), (19456, 0.5516), (20480, 0.5517), (21504, 0.5517), (22528, 0.5523), (23552, 0.5523), (24576, 0.5523), (25600, 0.5527), (26624, 0.5527), (27648, 0.5527), (28672, 0.5527), (29696, 0.5527), (30720, 0.5527), (31744, 0.5527), (32768, 0.5527)]

**Reasoning Efficiency Frontier for DeepSeek-R1-Distill-Qwen-7B**

[(0, 0.0872), (64, 0.0892), (128, 0.0977), (192, 0.1143), (256, 0.1512), (320, 0.1959), (384, 0.2522), (448, 0.2766), (512, 0.2896), (576, 0.2939), (640, 0.294), (704, 0.2944), (768, 0.2991), (832, 0.3155), (896, 0.3265), (960, 0.3418), (1024, 0.3889), (2048, 0.4885), (3072, 0.5345), (4096, 0.5742), (5120, 0.5926), (6144, 0.6128), (7168, 0.6293), (8192, 0.6486), (9216, 0.662), (10240, 0.6742), (11264, 0.6872), (12288, 0.6936), (13312, 0.6977), (14336, 0.7023), (15360, 0.7039), (16384, 0.7066), (17408, 0.7123), (18432, 0.7147), (19456, 0.7167), (20480, 0.7173), (21504, 0.7177), (22528, 0.7201), (23552, 0.7208), (24576, 0.7212), (25600, 0.7219), (26624, 0.7219), (27648, 0.7219), (28672, 0.7222), (29696, 0.7222), (30720, 0.7222), (31744, 0.7225), (32768, 0.7221)]

**Evaluating Reasoning Efficiency Gap.** To practically evaluate REG, instead of strictly following Eq. 5, we obtain approximations of $\sum_{L=1}^{L_{\max}} \hat{J}_{\text{optimal}}$ and $\sum_{L=1}^{L_{\max}} J(\mathcal{D}, \theta, L)$ through numerical integration on a set of token budgets respectively, in a similar way to Eq. 8. We select $\{L_1, \cdots, L_N\} = \{64i | 0 \le i < 16\} \cup \{1024i | 1 \le i \le 16\}$. Note that we set $L_{\max} = 16K$ instead of $L_{\max} = 32K$ to focus on the efficiency gap under lower token budgets.

## G   Additional Results

### G.1   More Results on Frontier LRMs

To enable a more comprehensive comparison as discussed in Sec 6.3, we also evaluated Claude Sonnet 3.7 under two different configurations. The detailed results are presented in Table 9. Row 2 shows the performance of Claude Sonnet 3.7 without the use of thinking mode, while Row 3 reflects its performance under the low reasoning effort mode (with a maximum token limit of 16,384 and 30% of the tokens allocated as budget tokens). Under both configurations, Claude Sonnet 3.7 produces significantly shorter responses, but this comes at the cost of a noticeable drop in accuracy.

| Model | Length | Accuracy |
|---|---|---|
| Claude Sonnet 3.7 (Thinking) | 17478.87 | 90.0% |
| Claude Sonnet 3.7 (Without Thinking) | 906.01 | 54.15% |
| Claude Sonnet 3.7 (Low Reasoning Effect) | 3491.14 | 67.62% |
| DeepSeek R1 | 5156.39 | 98.2% |
| Qwen3-4B | 8631.61 | 95.8% |
| Qwen3-8B | 8898.57 | 94.9% |
| Qwen3-32B | 7755.99 | 97.2% |
| Vanilla RL - 7B | 7732.11 | 93.5% |
| REO-RL (Exp) - 7B (ours) | 4524.02 | 93.5% |

Table 9: Comparing the response length and accuracy with frontier LRMs.

### G.2   Results on all Benchmarks

## Generalization to Other Tasks

To demonstrate the generalizability of REG and REO-RL, we conduct cross-domain evaluations on coding tasks using models trained solely on math data. Additionally, we apply the REG framework and REO-RL to coding by fine-tuning Qwen3-4B on DeepCoder data Luo et al. [2025b].

### 1. Cross-Domain Evaluation (Math → Coding)

We evaluate models trained on math on two coding benchmarks, CodeContests and LiveCodeBench. We construct cross-domain efficiency frontiers by truncating responses under different token budgets and allowing an extra 512-token budget for python code generation. We then compute REG based on the cross-domain efficiency frontiers. 8 responses are generated for evaluation for each method.

Even when trained only on math, REO-RL achieves strong generalization to coding, significantly improving efficiency (lower REG) while maintaining accuracy.

### 2. Applying the REG & REO-RL to Coding (Qwen3-4B)

For frontier construction, we run RL with different token budgets to estimate the efficiency frontier. Regarding REO-RL, we run REO-RL (Exp) using exponentially spacing token budgets. To obtain length-constrained rewards, we use the model to generate python codes from truncated responses with an additional generation budget of 512 tokens. REO-RL (Exp) reduces REG and response length by 51.14% and 32.49% on coding while maintaining the overall accuracy.

| Method | Accuracy (%) ↑ | Length ↓ | REG |
|---|---|---|---|
| Base LRM | 29.2 | 16757.1 | 2136.8 |
| Vanilla RL | 42.7 | 12829.1 | 1073.3 |
| REO-RL (Exp) - Coef=1 | $46.9_{\uparrow 4.2}$ | $10624.2_{\downarrow 17.2\%}$ | $30.0_{\downarrow 97.2\%}$ |
| REO-RL (Exp) | $42.9_{\uparrow 0.2}$ | $8780.3_{\downarrow 31.6\%}$ | $574.5_{\downarrow 46.5\%}$ |
| REO-RL (Oracle) | $42.5_{\downarrow 0.2}$ | $8443.8_{\downarrow 34.2\%}$ | $784.0_{\downarrow 26.9\%}$ |
| RL w. Token Budget=1K | $17.3_{\downarrow 25.4}$ | $1487.4_{\downarrow 88.4\%}$ | $3321.1_{\uparrow 209.4\%}$ |
| RL w. Token Budget=2K | $22.4_{\downarrow 20.3}$ | $2423.9_{\downarrow 81.1\%}$ | $2610.3_{\uparrow 143.2\%}$ |
| RL w. Token Budget=4K | $31.4_{\downarrow 11.4}$ | $3991.0_{\downarrow 68.9\%}$ | $1490.4_{\uparrow 38.9\%}$ |
| RL w. Len Group Norm. Rew. | $44.4_{\uparrow 1.7}$ | $10486.0_{\downarrow 18.3\%}$ | $689.8_{\downarrow 35.7\%}$ |
| RL w. Len-Harmonizing Rew. | $41.5_{\downarrow 1.2}$ | $9119.2_{\downarrow 28.9\%}$ | $767.2_{\downarrow 28.5\%}$ |
| MRT | $44.0_{\uparrow 1.3}$ | $10365.8_{\downarrow 19.2\%}$ | $610.1_{\downarrow 43.2\%}$ |
| SFT$_{\text{Shortest}}$ | $43.2_{\uparrow 0.5}$ | $12898.1_{\uparrow 0.5\%}$ | $1024.0_{\downarrow 4.6\%}$ |
| SFT$_{\text{TOPS}}$ | $41.8_{\downarrow 0.9}$ | $10823.9_{\downarrow 15.6\%}$ | $1057.7_{\downarrow 1.4\%}$ |
| SimPO$_{\text{DAST}}$ | $22.9_{\downarrow 19.8}$ | $14029.4_{\uparrow 9.4\%}$ | $2583.4_{\uparrow 140.7\%}$ |
| SimPO$_{\text{Shortest}}$ | $35.8_{\downarrow 6.9}$ | $7422.1_{\downarrow 42.1\%}$ | $1170.0_{\uparrow 9.0\%}$ |
| SimPO$_{\text{TOPS}}$ | $15.6_{\downarrow 27.1}$ | $2647.4_{\downarrow 79.4\%}$ | $3639.7_{\uparrow 239.1\%}$ |

Table 10: Results of 1.5B Models on AIME 2024

| Method | Accuracy (%) ↑ | Length ↓ | REG |
|---|---|---|---|
| Base LRM | 23.5 | 16577.6 | 906.7 |
| Vanilla RL | 28.3 | 14109.4 | 749.2 |
| REO-RL (Exp) - Coef=1 | $32.9_{\uparrow 4.6}$ | $10298.4_{\downarrow 27.0\%}$ | $250.5_{\downarrow 66.6\%}$ |
| REO-RL (Exp) | $31.9_{\uparrow 3.5}$ | $8630.5_{\downarrow 38.8\%}$ | $292.8_{\downarrow 60.9\%}$ |
| REO-RL (Oracle) | $34.7_{\uparrow 6.4}$ | $8288.7_{\downarrow 41.3\%}$ | $232.2_{\downarrow 69.0\%}$ |
| RL w. Token Budget=1K | $13.5_{\downarrow 14.8}$ | $1282.8_{\downarrow 90.9\%}$ | $1403.1_{\uparrow 87.3\%}$ |
| RL w. Token Budget=2K | $17.5_{\downarrow 10.8}$ | $1966.0_{\downarrow 86.1\%}$ | $1105.9_{\uparrow 47.6\%}$ |
| RL w. Token Budget=4K | $26.1_{\downarrow 2.2}$ | $3770.8_{\downarrow 73.3\%}$ | $474.5_{\downarrow 36.7\%}$ |
| RL w. Len Group Norm. Rew. | $30.7_{\uparrow 2.4}$ | $10599.6_{\downarrow 24.9\%}$ | $553.5_{\downarrow 26.1\%}$ |
| RL w. Len-Harmonizing Rew. | $31.6_{\uparrow 3.2}$ | $8113.9_{\downarrow 42.5\%}$ | $316.0_{\downarrow 57.8\%}$ |
| MRT | $31.8_{\uparrow 3.4}$ | $10496.2_{\downarrow 25.6\%}$ | $458.1_{\downarrow 38.9\%}$ |
| SFT$_{\text{Shortest}}$ | $29.9_{\uparrow 1.6}$ | $14221.8_{\uparrow 0.8\%}$ | $629.7_{\downarrow 16.0\%}$ |
| SFT$_{\text{TOPS}}$ | $30.7_{\uparrow 2.4}$ | $11720.5_{\downarrow 16.9\%}$ | $549.1_{\downarrow 26.7\%}$ |
| SimPO$_{\text{DAST}}$ | $19.4_{\downarrow 9.0}$ | $11161.7_{\downarrow 20.9\%}$ | $1008.3_{\uparrow 34.6\%}$ |
| SimPO$_{\text{Shortest}}$ | $26.5_{\downarrow 1.9}$ | $7380.9_{\downarrow 47.7\%}$ | $672.1_{\downarrow 10.3\%}$ |
| SimPO$_{\text{TOPS}}$ | $18.3_{\downarrow 10.0}$ | $2412.2_{\downarrow 82.9\%}$ | $1056.8_{\uparrow 41.1\%}$ |

Table 11: Results of 1.5B Models on AIME 2025

| Method | Accuracy (%) ↑ | Length ↓ | REG |
|---|---|---|---|
| Base LRM | 71.6 | 9956.8 | 1709.0 |
| Vanilla RL | 83.8 | 7968.0 | 1424.0 |
| REO-RL (Exp) - Coef=1 | $84.5_{\uparrow 0.7}$ | $4990.8_{\downarrow 37.4\%}$ | $648.5_{\downarrow 54.5\%}$ |
| REO-RL (Exp) | $83.2_{\downarrow 0.6}$ | $6205.6_{\downarrow 22.1\%}$ | $562.8_{\downarrow 60.5\%}$ |
| REO-RL (Oracle) | $84.4_{\uparrow 0.5}$ | $4626.6_{\downarrow 41.9\%}$ | $821.2_{\downarrow 42.3\%}$ |
| RL w. Token Budget=1K | $65.6_{\downarrow 18.2}$ | $1111.3_{\downarrow 86.1\%}$ | $1354.2_{\downarrow 4.9\%}$ |
| RL w. Token Budget=2K | $73.1_{\downarrow 10.7}$ | $1547.1_{\downarrow 80.6\%}$ | $827.8_{\downarrow 41.9\%}$ |
| RL w. Token Budget=4K | $80.9_{\downarrow 2.9}$ | $2561.2_{\downarrow 67.9\%}$ | $493.8_{\downarrow 65.3\%}$ |
| RL w. Len Group Norm. Rew. | $82.7_{\downarrow 1.1}$ | $5911.5_{\downarrow 25.8\%}$ | $1097.2_{\downarrow 22.9\%}$ |
| RL w. Len-Harmonizing Rew. | $85.1_{\uparrow 1.2}$ | $3993.2_{\downarrow 49.9\%}$ | $648.5_{\downarrow 54.5\%}$ |
| MRT | $84.6_{\uparrow 0.8}$ | $6233.6_{\downarrow 21.8\%}$ | $1051.8_{\downarrow 26.1\%}$ |
| SFT$_{Shortest}$ | $82.5_{\downarrow 1.3}$ | $7513.7_{\downarrow 5.7\%}$ | $1370.4_{\downarrow 3.8\%}$ |
| SFT$_{TOPS}$ | $82.7_{\downarrow 1.1}$ | $6454.1_{\downarrow 19.0\%}$ | $1283.5_{\downarrow 9.9\%}$ |
| SimPO$_{DAST}$ | $68.1_{\downarrow 15.7}$ | $5955.5_{\downarrow 25.3\%}$ | $1514.4_{\uparrow 6.4\%}$ |
| SimPO$_{Shortest}$ | $77.7_{\downarrow 6.2}$ | $4020.8_{\downarrow 49.5\%}$ | $1038.2_{\downarrow 27.1\%}$ |
| SimPO$_{TOPS}$ | $58.5_{\downarrow 25.3}$ | $1497.8_{\downarrow 81.2\%}$ | $2179.7_{\uparrow 53.1\%}$ |

Table 12: Results of 1.5B Models on AMC 2023

| Method | Accuracy (%) ↑ | Length ↓ | REG |
|---|---|---|---|
| Base LRM | 55.3 | 13062.1 | 2471.8 |
| Vanilla RL | 66.2 | 14264.7 | 1770.8 |
| REO-RL (Exp) | $64.0_{\downarrow 2.3}$ | $7671.5_{\downarrow 46.2\%}$ | $849.3_{\downarrow 52.0\%}$ |
| REO-RL (Oracle) | $63.9_{\downarrow 2.4}$ | $9348.6_{\downarrow 34.5\%}$ | $1015.6_{\downarrow 42.6\%}$ |
| RL w. Token Budget=1K | $26.7_{\downarrow 39.6}$ | $1242.4_{\downarrow 91.3\%}$ | $6119.5_{\uparrow 245.6\%}$ |
| RL w. Token Budget=2K | $36.5_{\downarrow 29.8}$ | $1929.6_{\downarrow 86.5\%}$ | $4372.7_{\uparrow 146.9\%}$ |
| RL w. Token Budget=4K | $48.3_{\downarrow 17.9}$ | $3480.9_{\downarrow 75.6\%}$ | $2458.7_{\uparrow 38.8\%}$ |
| RL w. Len Group Norm. Rew. | $64.2_{\downarrow 2.1}$ | $10608.1_{\downarrow 25.6\%}$ | $1353.4_{\downarrow 23.6\%}$ |
| RL w. Len-Harmonizing Rew. | $65.5_{\downarrow 0.7}$ | $9167.7_{\downarrow 35.7\%}$ | $1169.7_{\downarrow 33.9\%}$ |
| MRT | $66.1_{\downarrow 0.1}$ | $9210.6_{\downarrow 35.4\%}$ | $1051.8_{\downarrow 40.6\%}$ |
| SFT$_{Shortest}$ | $67.7_{\uparrow 1.5}$ | $13197.2_{\downarrow 7.5\%}$ | $1512.1_{\downarrow 14.6\%}$ |
| SFT$_{TOPS}$ | $66.0_{\downarrow 0.2}$ | $13204.1_{\downarrow 7.4\%}$ | $1609.1_{\downarrow 9.1\%}$ |
| SimPO$_{DAST}$ | $60.1_{\downarrow 6.1}$ | $7500.5_{\downarrow 47.4\%}$ | $1348.4_{\downarrow 23.9\%}$ |
| SimPO$_{Shortest}$ | $65.5_{\downarrow 0.7}$ | $9348.1_{\downarrow 34.5\%}$ | $1138.9_{\downarrow 35.7\%}$ |

Table 13: Results of 7B Models on AIME 2024

| Method | Accuracy (%) ↑ | Length ↓ | REG |
|---|---|---|---|
| Base LRM | 39.7 | 14241.9 | 2055.6 |
| Vanilla RL | 52.9 | 16305.3 | 1502.3 |
| REO-RL (Exp) | 48.8$_{\downarrow 4.2}$ | 8361.1$_{\downarrow 48.7\%}$ | 767.2$_{\downarrow 48.9\%}$ |
| REO-RL (Oracle) | 49.0$_{\downarrow 4.0}$ | 9189.6$_{\downarrow 43.6\%}$ | 628.2$_{\downarrow 58.2\%}$ |
| RL w. Token Budget=1K | 19.7$_{\downarrow 33.2}$ | 1168.0$_{\downarrow 92.8\%}$ | 4789.6$_{\uparrow 218.8\%}$ |
| RL w. Token Budget=2K | 27.6$_{\downarrow 25.3}$ | 1831.6$_{\downarrow 88.8\%}$ | 3354.8$_{\uparrow 123.3\%}$ |
| RL w. Token Budget=4K | 35.4$_{\downarrow 17.5}$ | 3345.2$_{\downarrow 79.5\%}$ | 2119.4$_{\uparrow 41.1\%}$ |
| RL w. Len Group Norm. Rew. | 50.8$_{\downarrow 2.1}$ | 11486.8$_{\downarrow 29.6\%}$ | 851.4$_{\downarrow 43.3\%}$ |
| RL w. Len-Harmonizing Rew. | 51.1$_{\downarrow 1.8}$ | 10394.6$_{\downarrow 36.2\%}$ | 881.5$_{\downarrow 41.3\%}$ |
| MRT | 50.5$_{\downarrow 2.4}$ | 10559.3$_{\downarrow 35.2\%}$ | 711.4$_{\downarrow 52.6\%}$ |
| SFT$_{\text{Shortest}}$ | 53.6$_{\uparrow 0.7}$ | 15208.6$_{\downarrow 6.7\%}$ | 1309.2$_{\downarrow 12.9\%}$ |
| SFT$_{\text{TOPS}}$ | 52.0$_{\downarrow 0.9}$ | 15170.8$_{\downarrow 7.0\%}$ | 1349.6$_{\downarrow 10.2\%}$ |
| SimPO$_{\text{DAST}}$ | 46.2$_{\downarrow 6.7}$ | 7916.9$_{\downarrow 51.4\%}$ | 1013.8$_{\downarrow 32.5\%}$ |
| SimPO$_{\text{Shortest}}$ | 50.8$_{\downarrow 2.1}$ | 10292.3$_{\downarrow 36.9\%}$ | 862.4$_{\downarrow 42.6\%}$ |

Table 14: Results of 7B Models on AIME 2025

| Method | Accuracy (%) ↑ | Length ↓ | REG |
|---|---|---|---|
| Base LRM | 90.9 | 6177.3 | 1652.7 |
| Vanilla RL | 93.9 | 7259.8 | 1841.5 |
| REO-RL (Exp) | 93.4$_{\downarrow 0.5}$ | 4144.6$_{\downarrow 42.9\%}$ | 1067.1$_{\downarrow 42.1\%}$ |
| REO-RL (Oracle) | 94.7$_{\uparrow 0.8}$ | 4444.1$_{\downarrow 38.8\%}$ | 960.9$_{\downarrow 47.8\%}$ |
| RL w. Token Budget=1K | 73.0$_{\downarrow 20.9}$ | 960.4$_{\downarrow 86.8\%}$ | 2098.1$_{\uparrow 13.9\%}$ |
| RL w. Token Budget=2K | 84.3$_{\downarrow 9.6}$ | 1427.5$_{\downarrow 80.3\%}$ | 906.7$_{\downarrow 50.8\%}$ |
| RL w. Token Budget=4K | 89.1$_{\downarrow 4.8}$ | 2109.2$_{\downarrow 70.9\%}$ | 708.5$_{\downarrow 61.5\%}$ |
| RL w. Len Group Norm. Rew. | 94.5$_{\uparrow 0.5}$ | 4788.9$_{\downarrow 34.0\%}$ | 1181.3$_{\downarrow 35.9\%}$ |
| RL w. Len-Harmonizing Rew. | 94.5$_{\uparrow 0.5}$ | 4307.9$_{\downarrow 40.7\%}$ | 1027.8$_{\downarrow 44.2\%}$ |
| MRT | 93.8$_{\downarrow 0.2}$ | 4986.6$_{\downarrow 31.3\%}$ | 1242.6$_{\downarrow 32.5\%}$ |
| SFT$_{\text{Shortest}}$ | 93.6$_{\downarrow 0.3}$ | 6728.2$_{\downarrow 7.3\%}$ | 1755.5$_{\downarrow 4.7\%}$ |
| SFT$_{\text{TOPS}}$ | 93.3$_{\downarrow 0.6}$ | 6272.5$_{\downarrow 13.6\%}$ | 1653.4$_{\downarrow 10.2\%}$ |
| SimPO$_{\text{DAST}}$ | 92.0$_{\downarrow 1.9}$ | 4051.7$_{\downarrow 44.2\%}$ | 1050.1$_{\downarrow 43.0\%}$ |
| SimPO$_{\text{Shortest}}$ | 93.4$_{\downarrow 0.5}$ | 4793.1$_{\downarrow 34.0\%}$ | 1276.0$_{\downarrow 30.7\%}$ |

Table 15: Results of 7B Models on AMC 2023

| Method | Accuracy (%) ↑ | Length ↓ | REG |
|---|---|---|---|
| Vanilla RL | 66.2 | 14264.7 | 1770.8 |
| Base LRM | 55.3 | 13062.1 | 2471.8 |
| REO-RL (Exp) - Coef=1 | $62.6_{\downarrow 3.6}$ | $7218.5_{\downarrow 49.4\%}$ | $756.5_{\downarrow 57.3\%}$ |
| REO-RL (Exp) - N=10 | $64.0_{\downarrow 2.3}$ | $9352.7_{\downarrow 34.4\%}$ | $1088.9_{\downarrow 38.5\%}$ |
| REO-RL (Linear) | $63.5_{\downarrow 2.7}$ | $8982.0_{\downarrow 37.0\%}$ | $1097.8_{\downarrow 38.0\%}$ |
| REO-RL (Question-Specific Oracle) | $63.9_{\downarrow 2.4}$ | $8407.8_{\downarrow 41.1\%}$ | $831.4_{\downarrow 53.0\%}$ |
| REO-RL (Oracle) | $63.9_{\downarrow 2.4}$ | $9348.6_{\downarrow 34.5\%}$ | $1015.6_{\downarrow 42.6\%}$ |
| REO-RL (Exp) | $64.0_{\downarrow 2.3}$ | $7671.5_{\downarrow 46.2\%}$ | $849.3_{\downarrow 52.0\%}$ |

Table 16: Results of Ablation Study on DeepSeek-R1-Distilled-Qwen-7B on AIME 2024

| Method | Accuracy (%) ↑ | Length ↓ | REG |
|---|---|---|---|
| Vanilla RL | 52.9 | 16305.3 | 1502.3 |
| Base LRM | 39.7 | 14241.9 | 2055.6 |
| REO-RL (Exp) - Coef=1 | $47.2_{\downarrow 5.7}$ | $7525.9_{\downarrow 53.8\%}$ | $611.1_{\downarrow 59.3\%}$ |
| REO-RL (Exp) - N=10 | $48.9_{\downarrow 4.1}$ | $9804.9_{\downarrow 39.9\%}$ | $927.0_{\downarrow 38.3\%}$ |
| REO-RL (Linear) | $49.9_{\downarrow 3.0}$ | $9001.6_{\downarrow 44.8\%}$ | $715.1_{\downarrow 52.4\%}$ |
| REO-RL (Question-Specific Oracle) | $51.4_{\downarrow 1.6}$ | $9298.5_{\downarrow 43.0\%}$ | $401.2_{\downarrow 73.3\%}$ |
| REO-RL (Oracle) | $49.0_{\downarrow 4.0}$ | $9189.6_{\downarrow 43.6\%}$ | $628.2_{\downarrow 58.2\%}$ |
| REO-RL (Exp) | $48.8_{\downarrow 4.2}$ | $8361.1_{\downarrow 48.7\%}$ | $767.2_{\downarrow 48.9\%}$ |

Table 17: Results of Ablation Study on DeepSeek-R1-Distilled-Qwen-7B on AIME 2025

| Method | Accuracy (%) ↑ | Length ↓ | REG |
|---|---|---|---|
| Vanilla RL | 93.9 | 7259.8 | 1841.5 |
| Base LRM | 90.9 | 6177.3 | 1652.7 |
| REO-RL (Exp) - Coef=1 | $93.1_{\downarrow 0.8}$ | $3736.4_{\downarrow 48.5\%}$ | $804.4_{\downarrow 56.3\%}$ |
| REO-RL (Exp) - N=10 | $94.1_{\uparrow 0.2}$ | $4779.6_{\downarrow 34.2\%}$ | $1027.3_{\downarrow 44.2\%}$ |
| REO-RL (Linear) | $93.9_{0.0}$ | $4290.9_{\downarrow 40.9\%}$ | $1093.9_{\downarrow 40.6\%}$ |
| REO-RL (Question-Specific Oracle) | $93.4_{\downarrow 0.5}$ | $4230.6_{\downarrow 41.7\%}$ | $970.6_{\downarrow 47.3\%}$ |
| REO-RL (Oracle) | $94.7_{\uparrow 0.8}$ | $4444.1_{\downarrow 38.8\%}$ | $960.9_{\downarrow 47.8\%}$ |
| REO-RL (Exp) | $93.4_{\downarrow 0.5}$ | $4144.6_{\downarrow 42.9\%}$ | $1067.1_{\downarrow 42.1\%}$ |

Table 18: Results of Ablation Study on DeepSeek-R1-Distilled-Qwen-7B on AMC 2023

Table 19: DeepSeek-R1-Distill-Qwen-1.5B Results (Trained on Math, Eval on Coding)

| Method | Acc. | Len. | REG |
|---|---|---|---|
| Vanilla RL | 13.55 | 15593.15 | 881.45 |
| REO-RL (Oracle) | 14.76 | 13064.89 | 556.57 |
| REO-RL (Exp) | 14.83 | 14047.32 | 510.56 |
| RL w. Len Group Norm. Rew. | 13.2 | 14328 | 996.29 |
| RL w. Len-Harmonizing Rew. | 14.34 | 12670.71 | 646.61 |
| RL w. Token Budget=4K | 13.35 | 6144.13 | 876.90 |

Table 20: DeepSeek-R1-Distill-Qwen-7B Results (Trained on Math, Eval on Coding)

| Method | Acc. | Len. | REG |
|---|---|---|---|
| Vanilla RL | 33.2 | 14722.48 | 849.27 |
| REO-RL (Oracle) | 32.29 | 10246.18 | 747.14 |
| REO-RL (Exp) | 31.98 | 9143.80 | 789.20 |
| RL w. Len Group Norm. Rew. | 32.48 | 13183.96 | 839.00 |
| RL w. Len-Harmonizing Rew. | 31.49 | 13240.02 | 1070.17 |
| RL w. Token Budget=4K | 29.40 | 4968.69 | 1353.91 |

Table 21: Qwen3-4B Results on Coding

| Method | Acc. | Length | REG |
|---|---|---|---|
| Base LRM (Qwen3-4B) | 41.14 | 16850.23 | 514.86 |
| REO-RL (Exp) | 41.03 | 11375.06 | 251.55 |
| RL w. Token Budget=8K | 28.78 | 7059.58 | 2572.08 |
| RL w. Token Budget=16K | 32.58 | 10443.31 | 1961.00 |

