# OpenReview forum: "How Far Are We from Optimal Reasoning Efficiency?"
_NeurIPS.cc/2025/Conference — NeurIPS 2025 poster_

### Official Review · Reviewer_uvtF · 2025-07-01

**Clarity:** 2
**Significance:** 2
**Originality:** 2
**Rating:** 3
**Confidence:** 4

**Summary:**

This paper addresses the overthinking behavior in reasoning models which often produce verbose outputs. The authors introduce Reasoning Efficiency Frontiers which is an empirical upper bounds on the accuracy-length trade-off, and define Reasoning Efficiency Gap (REG) to measure deviation from these frontiers. To optimize REG, they propose REO-RL, a reinforcement learning algorithm that approximates a dense multi-budget reward objective using a small set of token budgets selected via either oracle or exponential strategies. Experiments on three math reasoning benchmarks with 1.5B and 7B models show REO-RL improves reasoning efficiency over baselines.

**Questions:**

- Could authors include an algorithm pseudo code for REO-RL?
- Could authors provide more details on how the reasoning efficiency frontier is constructed? Does the author train RL with three different objectives, evaluate on each token budget, and select the best one over the three methods?
- Since exp with coef = 1 outperforms oracle method, I wonder what would be the actual use for such an empirical frontier.

**Ethical Concerns:**

["NO or VERY MINOR ethics concerns only"]

**Final Justification:**

The frontier estimation and training efficiency issues have been resolved. The estimated frontier helps to establish an exponential budget selection.

The issue remain unresolved is the supplied pseudo-code appears to be misaligned with the stated objective. According to Equation (6), we aim to train a model that performs optimally at every token budget (which optimally would be the red dashed line in Figure 2). Yet both the pseudo-code and Equation (9) train only on the full response y, but not on any truncated responses y_trunc. As a result, the model may encounter out-of-distribution issues when generating under smaller budgets, since it has never been trained to predict directly from y_trunc.

I hypothesize that the underlying objective of the algorithm is to encourage the model to generate important reasoning steps as early as possible. For example, the model generates two responses, one where it first interprets the question for 1k tokens and solves it for 1k tokens, and the other generation interprets and solves together in 2k tokens. Then, the second generation would have a higher chance that, if you truncate at 1k token budget, it will directly produce the right answer. Therefore, the model would learn to generate important reasoning steps as early as possible, which might be the reason that REO-RL is able to outperform other token efficiency algorithms. The current draft feels that there is a disconnection between the empirical frontier and the proposed algorithm, as the algorithm does not directly optimize to improve towards the frontier. Therefore, I have a final rating of 3.

**Limitations:**

yes

**Quality:**

2

**Strengths And Weaknesses:**

Strengths:
- The empirical results are strong, with comprehensive comparisons across baselines, model sizes, and ablations.
- The paper is well-structured.
- REG metric offers a useful lens for benchmarking reasoning efficiency.
- The idea of optimizing a sparse approximation to a dense budgeted reward objective is novel in this context.

Weaknesses:
- The entire framework relies on empirical frontiers derived from a limited set of fine-tuned models. There is no theoretical guarantee that these frontiers are close to optimal, which weakens the central premise.
- Prompting the model to generate the final answer directly from a truncated trace to be used for intermediate rewards is not justified, since the objective in Equation 2 is on $y$ sampled from $\pi$ rather than the truncated response. (Maybe the authors are actual training on the truncate trace for each truncation under different token budget. It would be helpful to include an algorithm pseudo code.)
- The complexity of the algorithm is higher than the baselines as it needs to regenerate for N times after each generation ends.
- The details on how the reasoning efficiency frontier is constructed are missing.
- Empirically the oracle method is close to exp with coef = 1. Exp with coef = 1 is essentially just optimizes the mean rewards for different token budgets which defeats the purpose of optimizing with an oracle frontier. The frontier does not seem to be useful except for being a metric.

---

> ### Author Rebuttal · Authors · 2025-07-31
>
> We sincerely appreciate the reviewer's time and feedback. Below, we address the key concerns point-by-point:
>
> 1. **Large-Scale Efficiency Frontier Estimation**
>     - To rigorously estimate optimal reasoning efficiency, we conducted **large-scale RL experiments** spanning **8 algorithms and 15 training configurations**, resulting in **180+ and 210+ models for 1.5B and 7B scales**, respectively. This far exceeds the limited fine-tuned models mentioned by the reviewer. This large-scale evaluation ensures robust, statistically meaningful efficiency frontiers.
> 2. **Training Efficiency of REO-RL**
>     - We provide an **empirical comparison** showing that *training efficiency of REO-RL is competitive with baselines.*
>     - We provide a **theoretical analysis** explaning why *REO-RL is training-efficient.*
> 3. **Are the the efficiency frontiers useful in our design of REO-RL?**
>     - It is **through the efficiency frontiers that we found using the exponential budget selection is principled in REO-RL.** (Sec 5.2). This further motivates us to propose REO-RL(Exp) ad REO-RL(Exp)-Coef=1 as practical implementations.
>     - We show via an ablation study that **exponential budget selection is critical for reasoning efficiency**. Please see our response to [W3,Q3] below.
>
> To summarize, our work introduces:
> - **Efficiency frontiers** as the first large-scale estimation of optimal reasoning efficiency.
> - **REG** as a practical metric for quantifying gaps to optimality.
> - **REO-RL(Exp)** as a *training-efficient* and *practical* algorithm, empirically and theoretically justified.
>
> We hope this clarifies our methodology and adequaetly addresses the reviewer's concern. We appreciate the reviewer's consideration and are happy to address any further questions.
>
> ## [W1,W4,Q2] Large-Scale Efficiency Frontier Estimation
>
> To ensure representative coverage of the best methods across various token budgets, we run RL training with **3 classes of algorithms**, covering **8 distinct algorithms** and **15 unique training configurations** in total. Each  individual run consumes at least 2K and 4K GPU hours for each run in the 1.5B and 7B scales, respectively. For each run, a set of checkpoints are sampled for evaluation. In total, **186 and 218 models in 1.5B and 7B settings**, respectively, are evaluated for constructing the efficiency frontiers. **Our large-scale experiments provide statistically meaningful frontiers that empirically approximate the optimal reasoning efficiency.**
>
> For each model $\pi_{\theta_i}$, we measure the accuracy across benchmarks under token budgets $\{32,64,\cdots,512,1K,2K,\cdots,32K\}$. Finally, the efficiency frontier for each benchmark is derived by selecting the **maximum accuracy across all models** for each token budget.
>
> The table below summarizes the algorithm classes, configurations, and model counts:
>
> | Algorithm Class         | Specific Algorithms & Training Configurations                                                               | # of Models (1.5B) | # of Models (7B) |
> |-------------------------|-------------------------------------------------------------------------------------|--------------------|------------------|
> | RL w. Token Budget      | Token Budget $\in \{512, 1K, 2K, 4K, 8K, 16K, 32K\}$                                    | 52                 | 118              |
> | RL w. Length Reward     | MRT; RL w. Len. Group Norm. Rew. ($\alpha\in\{0.1, 0.2\}$); RL w. Len.-Harmonizing Rew. ($\lambda \in \{1, 2\}$) | 83                 | 70               |
> | Preference Learning     | SimPO(Shortest, TOPS, DAST)        | 51                 | 30               |
> | Others | Base LRM, Vanilla RL | - | - |
>
>
> Further details about the algorithms and frontiers can be found in **Line 141-151** and **Appendix C**.
>
> ## [W3] Training Efficiency of REO-RL
>
> We thank the reviewer for raising concerns about training efficiency. Actually, REO-RL is training-efficient. Here we provide training efficiency comparison with baselines, and a theoretical analysis for the cost of REO-RL.
>
> To start with, REO-RL generates a response and then computes rewards by,
> 1. generate a response $y\sim\pi_\theta(\cdot|x)$
> 2. For each truncated response $y_{:L_i}$, an answer $a_i$ is generated, i.e. $a_i\sim \pi_\theta(\cdot|x, y_{:L_i},\text{[The Final Answer is]})$, with a budget of 10 tokens. Reward is then computed by matching the answer with the ground-truth answer.
>
> ### 1. Training Efficiency Comparison
>
>  Below, we provide empirical evidence showing that REO-RL is computationally efficient:
>
> (a) **Low Overhead from Answer Generation**
> REO-RL uses *a small number of token budgets $N=5$* and the *answers are short*, i.e. $|a_i|\le 10$. The per-question latency breakdown shows that **answer generation adds only 6.4% overhead relative to full response generation**:
>
> | Generating a response y | Generating Answers a[1]..a[N] |
> |---|---|
> |  110.63 $\pm$ 63.80 s| **7.53 $\pm$ 5.02s** |
>
> (b) **Faster Training Than Baselines**
>
> REO-RL trains faster than Vanilla RL and other RL methods with length rewards. This is because *overhead from answer generation is low and REO-RL  incentivizes shorter response length*.
>
> | Method                          | Avg. Time per Training Step (s) | Avg. Length (Test) | Avg. Acc. (Test) |
> |---------------------------------|------------------------|-------------|-----------|
> | Vanilla RL                      | 1147.39                | 12609.9     | 71.0      |
> | **REO-RL (EXP)**                 | **877.36**                 | 6725.7      | 68.7      |
> | **REO-RL (Oracle)**                 | **873.26**                 | 7660.8      | 69.2      |
> | RL w. Length Group Norm. Reward | 938.58                 | 8961.3      | 69.8      |
> | RL w. Length-Harmonizing Reward | 903.93                 | 7956.8      | 70.4      |
> | RL w. Token Budget=4K           | 400.13                 | 2978.4      | 57.6      |
>
> Specially, RL w. Token Budget excessively reduces length at the cost of severe accuracy drop,also having fast training speed.
>
> ### 2. Theoretical Cost Analysis
>
> Following [1] and [2], we analyze the computation cost for a single input. Let $L_{in}$ be the length of the input, $L_{out}$ be the length of generated response $y$ and  $L_{ans}$ be the maximum lengths of answers generated from truncated responses. For a transformer with $P$ parameters, KV size $D$ and GQA ratio $r$, the computation in REO-RL consists of three parts:
>
> - Prefill for Input $x$: $2PL_{in}+2DL_{in}^2$
> - Decoding Response $y$: $2PL_{out}+r(2L_{in}+L_{out})L_{out}D$
> - Decoding Answers $a_1,\cdots,a_N$: $2PNL_{ans}+r(2L_{in}+2L_{out}+L_{ans})NL_{ans}D$
>
> **Why REO-RL is Efficient**:
>
> 1. **Negligible Answer Generation Cost.** The input length $L_{in}$ is usually small ($\le 500$) and the response length $L_{out}$ is large, spanning from 5K to 32K. In REO-RL, by default we use $N=5$ and $L_{ans}\le 10$.  Since $L_{out}\gg NL_{answer}$, the computation cost is dominated by decoding $y$.
> 2. **KV Cache Reuse:** The KV cache of $x$ and $y$ could be reused during answer generation. For more details on KV cache, please refer to [SGLang Blog](https://lmsys.org/blog/2024-01-17-sglang/).
>
> [1] Kinetics: Rethinking Test-Time Scaling Laws. https://arxiv.org/pdf/2506.05333
>
> [2] Large Language Monkeys: Scaling Inference Compute with Repeated Sampling. https://arxiv.org/abs/2407.21787
>
> ## [W5,Q3] Utility of Frontiers in REO-RL & Novelty of Exponential Budget Selection
>
> We appreciate the reviewer's thoughtful comments questioning the utility of the frontiers.  Below, we clarify practical utility and the novelty of the frontiers:
>
> **1. In REO-RL, the frontiers reveal that exponential budget selection is principled for efficient reasoning.**
>
> As shown in Fig. 3 (Sec 5.2), oracle budget selection naturally follows an exponential spacing pattern. This further motivates exponential spacing as a principled and practical way to select token budgets. Thus, REO-RL (Exp) is not an arbitrary heuristic, but derived from the inherent structure of the frontiers.
>
> **2. Exponential budget selection is critical for reasoning efficiency**.
>
> We make a comparison with a linear variant of REO-RL (with Coef=1). Clearly using linear budget selection underperforms the exponential budget selection:
>
> |  | Accuracy | Length $\downarrow$ | REG $\downarrow$ |
> |-|-|-|-|
> | REO-RL(Exp) | 68.7 | 6725.7 | **816.7** |
> | REO-RL(Exp) - Coef=1 | 67.6 | 6160.3 | **605.9** |
> | REO-RL(Linear) - Coef=1 | 69.1 | 7424.8 | 886.1 |
>
>
> ## [W2, Q1] Pseudo-Code of REO-RL
>
> ```
> Input:
>     - LLM π
>     - Dataset D consisting of input prompts and ground-truth answers
>     - Token budgets L = [L[1], ..., L[N]]
>     - Training steps T
>
> Output: Fine-tuned LLM π*
>
> def Generate(π, x, ans, L):
>     y ~ π(· | x)  ▷ Generate full response
>     a = []  ▷ Stores answers for truncated responses
>
>     ▷ Generate truncated answers in parallel for budgets L[i] < |y|
>     for i = 1 to N do
>         if L[i] < |y| then
>             y_trunc = x + y[:L[i]] + "</think> The Final Answer is: "
>             a[i] ~ π(· | y_trunc)  ▷ Answer from truncated response
>         else:
>             a[i] = ExtractAnswer(y) ▷ Answer from the full response
>
>     ▷ Compute reward (Eq. (2) and Eq. (9) in paper)
>     reward = compute_reward(y, a, ans, L)
>
>     return reward, y
>
> for t = 1 to T do
>     ▷ Sample a batch of prompts and answers
>     B ← Sample a training batch from D
>     rewards, responses = [], []
>
>     ▷ Parallel rollout for each (x, ans) in B
>     for (x, ans) in B do
>         reward, response <- Generate(π, x, ans, L)
>
>     ▷ PPO update step
>     π ← Update with PPO
> end for
> return π
> ```
> Note that the responses and rewards are generated in parallel to ensure high training efficiency, known as async training in modern LLM training frameworks such as [AReaL](https://github.com/inclusionAI/AReaL) and [veRL](https://github.com/volcengine/verl/pull/2231).

---

> > ### Comment · Reviewer_uvtF · 2025-08-04
> >
> > Thank you for the detailed response. My concerns about frontier estimation and training efficiency have been resolved. I now understand how the estimated frontier helps to establish an exponential budget selection.
> >
> > That said, the supplied pseudo-code appears to be misaligned with our stated objective. According to Equation (6), we aim to train a model that performs optimally at every token budget (which optimally would be the red dashed line in Figure 2). Yet both the pseudo-code and Equation (9) train only on the full response y, but not on any truncated responses y_trunc. As a result, the model may encounter out-of-distribution issues when generating under smaller budgets, since it has never been trained to predict directly from y_trunc.
> >
> > I hypothesize that the underlying objective of the algorithm is to encourage the model to generate important reasoning steps as early as possible. For example, the model generates two responses, one where it first interprets the question for 1k tokens and solves it for 1k tokens, and the other generation interprets and solves together in 2k tokens. Then, the second generation would have a higher chance that, if you truncate at 1k token budget, it will directly produce the right answer. Therefore, the model would learn to generate important reasoning steps as early as possible, which might be the reason that REO-RL is able to outperform other token efficiency algorithms. The current draft feels that there is a disconnection between the empirical frontier and the proposed algorithm, as the algorithm does not directly optimize to improve towards the frontier.  Therefore, I will raise the rating to 3.

---

> ### Author Response · Authors · 2025-08-05
>
> We sincerely appreciate the reviewer’s thoughtful feedback and are happy to clarify the remaining points. Below, we address each concern in detail:
>
> 1.**REO-RL Optimizes Across All Token Budgets**
>
> The reviewer raises a point about Eq. (9) appearing to focus only on the full response $y$. However, **REO-RL inherently optimizes performance at every token budget, as rewards are computed from truncated responses**. Crucially, when $N=L_{max}-1$ and $L_i=i,\forall i$, the REO-RL objective (Eq. 9) reduces exactly to the full efficiency objective (Eq. 6):
>
> $
> L_{REO-RL}(\theta,D)=E_{x\sim D}[E_{y\sim\pi_\theta(\cdot|x)}[\sum_{i=1}^{Lmax}\frac{1}{2}R(x,\text{Answer}(\pi_\theta, x,y_{:i}))]]
> $
>
> $
> \quad\quad\quad\quad\quad\quad=\frac{1}{2}\sum_{i=1}^{Lmax}E_{x\sim D}[E_{y\sim\pi_\theta(\cdot|x)}[R(x,\text{Answer}(\pi_\theta, x,y_{:i}))]]
> $
>
> $
> \quad\quad\quad\quad\quad\quad=\frac{1}{2}\sum_{i=1}^{Lmax}J(D,\theta,i) =\frac{1}{2} L_{efficiency}(D,\theta)
> $
>
> This shows that REO-RL is a practical approximation of optimizing across all budgets. As demonstrated in Sec 5.2 and Figure 3, with $N\ge 5$, the approximation error is negligible, meaning REO-RL effectively improves performance across all budgets.
>
> 2. **Answer Generation from Truncated Responses**
>
> The reviewer raises a valid concern about whether the model is trained to generate answers directly from truncated responses $y_{trunc}$. We clarify that explicit training for this is unnecessary because the model naturally generates well-formatted answers when prompted with:
>
> > *$y_{trunc}$ + ... The Final Answer is \boxed{*
>
> In practice, with a small additional budget (e.g., 10 tokens), the model reliably produces the best possible answer without further reasoning. This approach is simple yet effective, avoiding the need for additional training complexity.
>
> 3. **Connection Between Frontiers and REO-RL**
>
> The reviewer notes a perceived disconnection between the empirical frontier and REO-RL. We emphasize that:
>
> - **Frontier -> Full Efficiency Objective (Eq. 6)**: Minimizing the REG metric (Eq. 5) is equivalent to maximizing the full efficiency objective (Eq. 6), which pushes the model toward the frontier.
>
> - **Full Efficiency Objective (Eq. 6) -> REO-RL (Eq. 9)**: REO-RL (Eq. 9) is a tractable approximation of Eq. 6, converging to it as $N$ increases.
>
> We will revise the manuscript to explicitly link these steps (REG (Eq.5) → full efficiency objective (Eq. 6) →  REO-RL (Eq. 9)), making the connection clearer. These revisions would be made in Line 170-`174 and Line 187-191.
>
> 4. **Broader Contributions**
>
> Finally, we highlight that REO-RL is just one part of our contribution. Beyond REO-RL, our key advancements include:
> - The first large-scale estimation of optimal reasoning efficiency.
> - The REG metric which provides practical measurement of the gap to the optimality.
>     These provide a foundation for future work beyond our proposed algorithm.
>
> We appreciate the reviewer’s thoughtful comments and hope these clarifications resolve the concerns.

---

> > ### Comment · Reviewer_uvtF · 2025-08-05
> >
> > Thank you for the quick reply.
> >
> > 1. I understand that rewards are computed at every token budget. However, during model training, gradient descent is applied **only** to the full responses. This may lead to a distributional shift between training and testing, as the model is never **explicitly** trained to produce final answers based on truncated reasoning traces.
> >
> > 2 and 3. I agree that modern LLMs are capable of generating well-formatted answers even from truncated responses, and I also see that the frontier is constructed based on these truncated outputs at each token budget. That said, there appears to be a disconnect between how the frontier is defined and how the model is trained. Again, the model is not directly trained to answer questions based on truncated traces. Instead, training still relies on full responses with modified rewards, which encourages the model to generate important reasoning steps as early as possible. While the frontier is constructed based on truncated outputs at each token budget.
> >
> > I will maintain my current score. Thank you again for the clarifications.

---

> > > ### Author Response · Authors · 2025-08-07
> > >
> > > We sincerely appreciate the reviewer’s thoughtful feedback and previous time on discussion. Below, we address the remaining concerns with new experiments, empirical evidence, and clarifications:
> > >
> > > ### 1. **New Experiments: Directly Optimizing Answer Prediction from Truncated Responses**
> > >
> > > To explicitly test the impact of optimizing the accuracy of answer generation from truncated responses, we conducted an ablation study augmenting REO-RL (Exp) with an additional PPO objective to improve answer generation from truncated traces. We name this variant **REO-RL (Exp) w. Aug.**
> > >
> > > Results:
> > > - **Minimal Accuracy Gain:** Improving accuracy from truncated responses yields negligible improvement (e.g., +0.4% for 1.5B, -0.7% for 7B).
> > >
> > > - **No Consistent Efficiency Benefit**: Efficiency (REG) is marginally better for 1.5B but slightly worse for 7B, suggesting no clear benefit for training the model to directly predict from truncated responses.
> > >
> > > *DeepSeek-R1-Distill-Qwen-1.5B Result:*
> > >
> > > | Method | Accuracy | Length | REG |
> > > |-------------|-|-|-------------------|
> > > | Vanilla RL | 51.6 | 11635.5 | 1257.0 |
> > > | REO-RL (Exp) | 53.1 | 7467.2 | 551.8 |
> > > | REO-RL (Exp) w. Aug | 53.5 | 7283.4 | 537.9 |
> > > | RL w. Token Budget=4K | 46.1 | 3441.0  | 1045.3 |
> > >
> > >
> > > *DeepSeek-R1-Distill-Qwen-7B Result:*
> > >
> > > | Method | Accuracy | Length | REG |
> > > |-------------|-|-|-------------------|
> > > | Vanilla RL | 71.0 | 12609.9 | 1691.6 |
> > > | REO-RL (Exp) | 68.7 | 6725.7 | 816.7 |
> > > | REO-RL (Exp) w. Aug | 68.0 | 6530.4 | 847.3 |
> > > | RL w. Token Budget=4K | 57.6 | 2978.4  | 1343.3 |
> > >
> > > **Why No Significant Improvement?**
> > >
> > > We hypothesize that generating answers (e.g., 10 tokens) from truncated traces offers limited room for improvement, as reasoning quality fundamentally depends on full traces (often thousands of tokens). While the reviewer’s concern is theoretically valid, empirical results suggest this direction may not meaningfully advance efficiency. We thank the reviewer for raising this point and will include these findings in our revision.
> > >
> > > ### 2. **Empirical Envidences Already Show REO-RL Improves the Model Towards the Frontiers**
> > > As shown in Sec. 6.2 (Fig. 4, Table 1), REO-RL consistently improves accuracy across token budgets and reduces REG versus the base LRMs and vanilla RL (1.5B and 7B scales). This confirms that training on full responses inherently improves the model closer to the frontiers, even without explicit optimization for answer prediction from truncated traces.
> > >
> > > ### 3. **How REO-RL Encourages the Model to Align Better with Frontiers**
> > > Our observations indicate that REO-RL empirically induces two key behaviors that align with efficiency frontiers:
> > > - **Reduced Redundancy**: The model learns to omit trivial reflections.
> > > - **Early Key Steps with Preserved Reflections**: Critical reasoning appears earlier without sacrificing correctness. *Importantly, the model still performs reflection over the key steps* to ensure correctness.
> > >
> > > Both behaviors empirically improve efficiency, with lower REG and higher accuracy across different budgets, suggesting closer alignment with the frontiers.
> > >
> > > **Clarifying the Reviewer’s Concern:**
> > >
> > > If we understand correctly, the reviewer suggests a concern on whether generating important steps early conflicts with closer alignment with the frontiers. We argue that this is *not a conflict but rather a mechanism for improving efficiency*:
> > > - The frontier is constructed from truncated responses, and REO-RL directly optimizes for the length-constrained rewards across all token budgets. Generating key steps early is a  part of the mechanism how REO-RL improves reasoning efficiency.
> > > - We emphasize that **the optimal reasoning behaviors for reasoning efficiency still remains an open problem**. Our empirical results robustly support that *REO-RL training effectively approaches the frontiers*.
> > >
> > > We welcome further discussion on this point and would value the reviewer’s thoughts on why they see early key steps as incompatible with frontier alignment.
> > >
> > > ### 4. **Broader Contributions**
> > >
> > > Beyond algorithmic details, our work provides:
> > >
> > > - **Efficiency frontiers** as the first large-scale estimation for optimal reasoning efficiency.
> > > - **REG Metric** to quantify gaps to optimality.
> > > - **Practical Algorithm**(REO-RL (Exp)) that outperforms baselines across all model sizes (Sec. 6.2).
> > >
> > > We appreciate the reviewer’s engagement and hope these clarifications address the remaining concerns.

---

> > > > ### Comment · Reviewer_uvtF · 2025-08-07
> > > >
> > > > Thank you for the response and for providing the additional experiments.
> > > >
> > > > However, the new results raise some concerns. In particular, REO-RL (Exp) w. Aug is intended to directly optimize toward the estimated frontier. The fact that REO-RL (Exp), which does not use this augmentation, performs comparably suggests that optimizing toward the frontier may not be as effective as intended. This weakens the motivation for introducing the frontier-based objective.
> > > >
> > > > To clarify the algorithmic issue more explicitly: the original REO-RL objective only considers the full response, which is misaligned with the reward structure---rewards are computed based on truncated responses at each token budget. This means that optimizing Equation (8) is not equivalent to optimizing Equation (9), nor can it be interpreted as a standard RL objective with dense rewards. Therefore, the statement on line 185 appears to be incorrect.
> > > >
> > > > For example, consider a full response $y_{1:N}$, where $y_{1:n}$ denotes the truncated response at token budget $L_n$. Each of these truncated responses can yield an answer $a_1, \dots, a_N$. The resulting trajectory is a skewed binary tree: the root node is $y_1$, which branches into $y_2$ and $a_1$, and $y_2$ further branches into $y_3$ and $a_2$, and so on. Each answer $a_n$ receives a reward. It is unclear how PPO would be applied to such a trajectory structure, and I am not sure that REO-RL (Exp) w. Aug accurately captures or optimizes this structure.
> > > >
> > > > On the other hand, even if we ignore the answers and treat it as a RL problem with dense rewards under one trajectory $y_{1:N}$, the algorithm in the pseudo-code still does not optimize the correct reward allocation. The dense rewards are allocated at each token budget, therefore the value for each step would be the sum of the rewards at all future steps. However, the algorithm in the pseudo-code treats step level reward as terminal reward.
> > > >
> > > > Given the theoretical disconnections between the REO-RL objective and the true structure of the frontier-based optimization, I am concerned that the current form may not meet the bar for NeurIPS.

---

> > > > > ### Author Response · Authors · 2025-08-08
> > > > >
> > > > > Thank you for your in-depth discussion. Below, we respond to your main points.
> > > > >
> > > > > **Concern 1: Dense vs. Terminal Rewards**
> > > > >
> > > > > In practice we use dense reward for REO-RL. Please find our first point **"Dense-Reward RL"** for more details. We also agree that Eq. (9) could be misleading and we have improved by directly presenting the policy gradient in the revision.
> > > > >
> > > > > **Concern 2: REO-RL does not directly optimize Eq. (8)**
> > > > >
> > > > > We here provide more details on REO-RL w. Aug, which is theorectically derived by computing the full gradient of Eq. (8). Please find our second point for more details. We highlight two key conclusions,
> > > > > 1. **Theoretically, optimizing towards the frontiers is a joint effect of improving truncated resoning (through REO-RL) and improving answer generation accuracy from truncated responses (through the augmentation)**
> > > > > 2. In practice, our ablation study empirically show **the mainly functioning part in optimizing towards the frontiers is improving truncated resoning through REO-RL.** We hypothesize that this is because answer generation accuracy from truncated responses has limited improvement space due to its limited length (10 answer tokens vs. >1k reasoning tokens).
> > > > >
> > > > > ## 1. Dense-Reward RL
> > > > >
> > > > > We clarify that the returned `reward` in the pseudo-code is an torch tensor with $|y|$ scalrs, which indeed provides dense rewards.
> > > > >
> > > > > In REO-RL, suppose we use REINFORCE algorithm (in practice we use PPO), the policy gradient is,
> > > > >
> > > > > $
> > > > > \nabla_\theta \mathcal L_{REO-RL}(\theta, \mathcal D)=\mathbb E_{x}[\mathbb E_{y\sim \pi_{\theta'}(\cdot|x,y)}[\sum_{i=1}^{N+1}\nabla_\theta\log \pi_{\theta}(y_{L_{i-1}:L_i}|x,y_{:L_{i-1}})\cdot(\sum_{j=i}^{N+1}c_j\mathcal R(x,\text{Answer}(\pi_{\theta'},x,y_{:L_j})))]]
> > > > > $
> > > > >
> > > > > where $\pi'$ is the last-iteration policy. This policy gradient follows standard dense-reward RL. The implementation details of REO-RL can also be found in Appendix D, line 545 - lin 553.
> > > > >
> > > > > ## 2. REO-RL w. Aug Implements Full Objective Optimization
> > > > >
> > > > > REO-RL (Exp) w. Aug is derived by theoretically analyzing the full gradient of Eq. (8):
> > > > >
> > > > > $\nabla_\theta f(\mathcal D, \theta,\{L_1,\cdots,L_N\})=\mathbb E_x[\sum_{i=0}^{N+1}c_i\nabla_\theta J(\mathcal D,\theta,L_i)]$
> > > > >
> > > > > where $c_i=\frac{L_{i+1}-L_{i-1}}{2},\forall 1\le i\le N$ and $c_{N+1}=\frac{Lmax-L_N}{2}, c_0=\frac{L_1}{2}$.
> > > > >
> > > > > The policy gradient for each token budget $L_i$ is,
> > > > >
> > > > > $\nabla_\theta J(\mathcal D,\theta, L_i)=\nabla_\theta[\mathbb E_{x\sim D}[\mathbb E_{y\sim \pi_\theta(\cdot|x)}\mathcal R(x, \text{Answer}(\pi_\theta, x, y_{:L_i}))]]$
> > > > >
> > > > > $=\mathbb E_{x\sim D}[\nabla_\theta[\mathbb E_{y\sim \pi_\theta(\cdot|x)}\mathcal R(x, \text{Answer}(\pi_{\theta'}, x, y_{:L_i}))]] + \mathbb E_{x\sim D}[\mathbb E_{y\sim \pi_{\theta'}(\cdot|x)}\nabla_\theta\mathcal R(x, \text{Answer}(\pi_{\theta}, x, y_{:L_i}))]$
> > > > >
> > > > > where $\theta'$ is equivalent to $\theta$ but does not propagate gradient. The second term is what the reviewer mostly concerns, i.e. optimizing the model to predict answers.
> > > > >
> > > > > The answer-prediction term can be further expanded to standard policy gradient,
> > > > >
> > > > > $\mathbb E_{x\sim D}[\mathbb E_{y\sim \pi_{\theta'}(\cdot|x)}\nabla_\theta\mathcal R(x, \text{Answer}(\pi_{\theta}, x, y_{:L_i}))]=\mathbb E_{x\sim D}[\mathbb E_{y\sim \pi_{\theta'}(\cdot|x)}[\mathbb I[|y|>L_i]\cdot \nabla_\theta\mathbb E_{a\sim\pi_\theta(\cdot|x,y_{:L_i},\text{[The Final Answer is]})} [\mathcal R(x, a)]]]$
> > > > >
> > > > > **Policy Gradient of REO-RL w. Aug:** Suppose we use REINFORCE algorithm for conciseness. The policy gradient of REO-RL w. Aug sums up two terms,
> > > > >
> > > > > - The first term is exactly the same as REO-RL:
> > > > > $
> > > > > \mathbb E_{x}[\mathbb E_{y\sim \pi_{\theta'}(\cdot|x,y)}[\sum_{i=1}^{N+1}\nabla_\theta\log \pi_{\theta}(y_{L_{i-1}:L_i}|x,y_{:L_{i-1}})\cdot(\sum_{j=i}^{N+1}c_j\mathcal R(x,\text{Answer}(\pi_{\theta'},x,y_{:L_j})))]]
> > > > > $
> > > > >
> > > > > - The second term optimizes answer generation accuracy from truncated responses. This is also the augmentation used in REO-RL w. Aug.
> > > > > $
> > > > > \mathbb E_{x}[\mathbb E_{y\sim \pi_{\theta'}(\cdot|x,y)}[\sum_{i=0}^{N+1}\mathbb I[|y|>L_i]\cdot \mathbb E_{a\sim \pi_{\theta'}(\cdot|x,y,\text{[The Final Ansewr is]})}[\nabla_\theta\log\pi_{\theta}(\cdot|x,y_{:L_i},\text{[The Final Ansewr is]})\cdot\mathcal R(x,a)]]]
> > > > > $
> > > > >
> > > > > Combining these two terms **exactly implements full optimization as questioned by the reviewer**.
> > > > >
> > > > > However, our ablation study suggests that improving answer generation from truncated responses brings no benefit over REO-RL. Our experiment results show that **improving truncated reasoning through REO-RL alone can effectively improve the models towards the frontiers.**
> > > > >
> > > > > ## Further Questions
> > > > >
> > > > > Given that it is the last-day of the discussion period, to ensure we address any remaining concerns, we’d appreciate your perspective on:
> > > > >
> > > > > - **Answer-Generation Optimization:** Why might you expect explicit training on truncated answers to help? For instance, do you hypothesize specific behaviors that would be enable?
> > > > >
> > > > > Thank you again for your time!

---

> > > > > > ### Comment · Reviewer_uvtF · 2025-08-08
> > > > > >
> > > > > > Thank you for your detailed feedback, and my apologies for overlooking the details in Appendix D. I agree that REO-RL w. Aug is optimizing the true objective. After reviewing the rebuttal and rereading the submission carefully, I believe a slightly more intuitive way to structure the paper could be:
> > > > > > 1. Present the empirical frontiers which are computed based on full and truncated responses.
> > > > > > 2. Propose REO-RL w. Aug which directly optimize the objective for the frontier.
> > > > > > 3. Propose REO-RL which is a simplified version of REO-RL w. Aug that achieves similar performance (which is a really novel contribution itself since the algorithm is much more simple now comparing to REO-RL w. Aug).
> > > > > >
> > > > > > I hope this suggestion does not come across as nitpicking. The reason I have repeatedly mentioned that the current flow feels somewhat disconnected is that the paper moves directly from step 1 to step 3, without explicitly including step 2. This naturally raises the question of how REO-RL w. Aug would perform if it were optimized directly toward the frontier. I will leave it to the AC to make the final judgment, as the current flow of the paper feels subjectively less intuitive to me.

---

> > > > > > > ### Author Response · Authors · 2025-08-09
> > > > > > >
> > > > > > > We scencerely thank the reviewer for the thorough feedback on our algorithm. We greatly appreciate the opportunity to address all raised concerns, and we agree that the in-depth discussion has helped us identify an important missed detail in the presentation of our main algorithm.
> > > > > > >
> > > > > > > To incorporate the reviewer’s suggestions, we will make the following revisions to the manuscript:
> > > > > > > 1. We will first discuss the full optimization of the objective in Eq. (9) (Line 185) before introducing the REO-RL objective. This will better illustrate how we simplify the full optimization problem into a more practical and efficient RL algorithm.
> > > > > > > 2. We will update Eq. (9) to explicitly include the policy gradient objective, making it clearer that REO-RL performs dense-reward optimization.
> > > > > > > 3. We will include the comparison between the full optimization of Eq. (9) and REO-RL to empirically justify our design choices.
> > > > > > >
> > > > > > > We believe these changes will significantly improve readability and better highlight the theoretical connection between the frontiers and the practical REO-RL algorithm
> > > > > > >
> > > > > > > Beyond the algorithmic contributions, our work also introduces:
> > > > > > > - **Efficiency frontiers**: the first large-scale estimation of optimal reasoning efficiency.
> > > > > > > - **REG Metric**: A quantitative measure for evaluating gaps to optimality.
> > > > > > >
> > > > > > > We hope these combined contributions provide valuable insights for future research on reasoning efficiency.
> > > > > > >
> > > > > > > Once again, we deeply appreciate the reviewer’s time and thoughtful feedback, which has undoubtedly strengthened our paper.

---

### Official Review · Reviewer_7ETN · 2025-07-02

**Clarity:** 2
**Significance:** 3
**Originality:** 2
**Rating:** 4
**Confidence:** 4

**Summary:**

This paper tackles the problem of verbose reasoning in Large Reasoning Models (LRMs). It first introduces Reasoning Efficiency Frontiers to define the empirical upper bound of the accuracy-length trade-off for any token budget. It then proposes the Reasoning Efficiency Gap (REG), a single metric that quantifies a model's deviation from this frontier. The core contribution is REO-RL, a novel reinforcement learning algorithm designed to minimize this gap. By leveraging sparse token budget sampling and numerical integration, REO-RL efficiently approximates the full-budget optimization problem, leading to substantial reductions in computational cost.

**Questions:**

The paper states on lines 136-137 that you "construct an empirical reasoning efficiency frontier by fine-tuning a diverse set of models using existing approaches." Could you please provide a more detailed explanation of how this frontier, which is used to estimate J_optimal, is constructed?

**Ethical Concerns:**

["NO or VERY MINOR ethics concerns only"]

**Final Justification:**

The author's supplementary experiments have addressed some key shortcomings of the paper to some extent, and I will raise my rating accordingly. Considering the maturity and completeness of the paper, the author needs to make significant adjustments to the experimental section in the final submission version.

**Limitations:**

yes

**Quality:**

3

**Strengths And Weaknesses:**

Strengths
1. The proposed REG metric effectively unifies model accuracy and efficiency into a single, coherent evaluation standard. The REO-RL algorithm, through sparse budget sampling, offers a computationally efficient optimization method.
2. The work demonstrates a high degree of experimental rigor, supported by a detailed appendix and the open-sourcing of code and data to ensure reproducibility.
3. The paper is clearly written with a smooth logical flow, and its core arguments are well-supported by intuitive figures and diagrams.

Weakness
1. The method's effectiveness is only demonstrated on mathematical reasoning tasks (AMC/AIME). Its applicability to other domains, such as programming, remains unverified, and the paper should either include cross-domain experiments or discuss the transfer potential of the REG framework.
2. Experiments are limited to relatively small models (1.5B/7B), and the paper does not assess the scalability of REO-RL on larger models (>10B). An analysis of its performance on larger-scale LRMs is necessary to confirm its broader utility.  My primary concern with this paper lies in the choice of models used for experimentation. The study relies on 1.5B and 7B models, which have limited practical value in real-world applications and are not representative of the state-of-the-art Large Reasoning Models currently in use. It is well-understood that the reasoning processes and emergent capabilities of models at this smaller scale can differ substantially from those in the 30B to 230B parameter range.
3. The paper does not quantify the additional computational overhead introduced by REO-RL, which requires generating multiple truncated responses for reward calculation. A comparative analysis of training efficiency is needed to provide a complete picture of the method's cost.

---

> ### Author Rebuttal · Authors · 2025-07-31
>
> We sincerely appreciate the reviewer’s time and thoughtful feedback, which has helped us strengthen our work. Below, we address the key concerns raised:
>
> 1. **Cross-Domain Generalization (Coding Tasks)**
>     - We evaluate the math-trained models on coding tasks to show the generalizability of REO-RL.
>     - We further include additional results **applying the REG framework and REO-RL to Qwen3-4B on coding** to further demonstrate the adaptability across domains.
> 2. **Scalability to Larger Models**
>     - To address the lack of evaluation on advanced large reasoning models, we **apply REO-RL to Qwen3-32B**, showing consistent improvements in efficiency-performance trade-offs
> 3. **Training Efficiency of REO-RL**
>     - We provide an **empirical comparison** showing that *training efficiency of REO-RL is competitive with baselines.*
>     - We provide a **theoretical analysis** showing why *REO-RL is training-efficient.*
> 4. **Efficiency Frontier Details**
>     - We included dedicated description including the specific training algorithms and training configuraitons, and construction method for the frontiers.
>
> We hope this clarifies our methodology and adequately addresses the reviewer's concerns. We’re grateful for the opportunity to improve our work and welcome further discussion.
>
> ## [W1] Cross-Domain Generalization (Coding Tasks)
>
>
> To demonstrate the **generalizability of REG and REO-RL**, we conduct **cross-domain evaluations on coding tasks** using models trained solely on math data. Additionally, we apply the **REG framework and REO-RL to coding** by fine-tuning **Qwen3-4B** on DeepCoder data[1].
>
> **1. Cross-Domain Evaluation (Math -> Coding)**
>
> We evaluate models trained on math (as in the paper) on two coding benchmarks, CodeContests and LiveCodeBench. We construct **cross-domain efficiency frontiers** by truncating responses under different token budgets and allowing an extra 512-token budget for code generation. We then compute REG based on the cross-domain efficiency frontiers. 8 responses are generated for evaluation.
>
> Though only trained on math, **REO-RL achieves strong generalization to coding**, significantly improving efficiency (lower REG) while maintaining accuracy.
>
> DeepSeek-R1-Distill-Qwen-1.5B:
>
> |Method|Acc.|Len.$\downarrow$|REG$\downarrow$|
> |----|-|-|-|
> |Vanilla RL|13.55|15593.15|881.45|
> |**REO-RL (Oracle)**|14.76|13064.89|**556.57**|
> |**REO-RL (Exp)**|14.83| 14047.32 |**510.56**|
> |RL w. Len Group Norm. Rew.|13.2 |14328|996.29|
> |RL w. Len-Harmonizing Rew.|14.34|12670.71|646.61|
> |RL w. Token Budet=4K|13.35|6144.13|876.90|
>
> DeepSeek-R1-Distill-Qwen-7B:
> |Method|Acc.|Len.$\downarrow$|REG$\downarrow$|
> |----|-|-|-|
> |Vanilla RL|33.2|14722.48|849.27|
> |**REO-RL (Oracle)**|32.29|10246.18|**747.14**|
> |**REO-RL (Exp)**|31.98|9143.80|**789.20**|
> |RL w. Len Group Norm. Rew.|32.48|13183.96|839.00|
> |RL w. Len-Harmonizing Rew.|31.49|13240.02|1070.17|
> |RL w. Token Budet=4K|29.40|4968.69|1353.91|
>
>
> **2. Applying the REG & REO-RL to Coding (Qwen3-4B)**
>
> For frontier construction, due to time constraints, we run **RL with different token budgets** to estimate the efficiency frontier.
>
> Regarding REO-RL, we run REO-RL (Exp) using *$N=5$ exponentially spacing token budgets*. To obtain length-constrained rewards, we use the model to generate codes from truncated responses with an addtional *generation budget of 512 tokens*. **REO-RL (Exp) reduces REG and response length by 51.14% and 32.49% on coding** while maintaining the overall accuracy.
>
> *Qwen3-4B Results on Coding:*
> |Method|Acc.|Len.$\downarrow$|REG$\downarrow$|
> |-|-|-|-|
> |Base LRM (Qwen3-4B)|41.14|16850.23|514.86|
> |REO-RL (Exp)|41.03|11375.06|**251.55**|
> |RL w. Token Budget=8K|28.78|7059.58|2572.08|
> |RL w. Token Budget=16K|32.58|10443.31|1961.00|
>
> [1] DeepCoder: A Fully Open-Source 14B Coder at O3-mini Level. https://huggingface.co/agentica-org/DeepCoder-14B-Preview
>
> ## [W2] Scalability to Larger Models
>
> We apply REO-RL (Exp) to an advanced LRM, Qwen3-32B using N=5 token budgets for training. Each model is evaluated on AIME24/AIME25/AMC23 with 32 responses. The results show that **REO-RL can reduce response length of Qwen3-32B by 10.03% while maintaining the accuracy**,indicating that *REO-RL can effectively scale to larger models*.
>
> ||Acc.|Length|
> |-|-|-|
> |Qwen3-32B|82.83|11961.23|
> |REO-RL (Exp)|83.47|10757.33|
>
> ## [W3] Computational Cost Analysis of REO-RL
>
> We thank the reviewer for raising concerns about training efficiency. Here we provide training efficiency comparison with baselines, and a theoretical analysis for the cost of REO-RL.
>
> To start with, REO-RL generates a response and then computes rewards by,
> 1. generate a response $y\sim\pi_\theta(\cdot|x)$
> 2. For each truncated response $y_{:L_i}$, an answer $a_i$ is generated, i.e. $a_i\sim \pi_\theta(\cdot|x, y_{:L_i},\text{[The Final Answer is]})$, with a budget of 10 tokens. Reward is then computed by matching the answer with the ground-truth answer.
>
> ### 1. Training Efficiency Comparison
>
>  Below, we provide empirical evidence showing that REO-RL is computationally efficient:
>
> (a) **Low Overhead from Answer Generation**
> REO-RL uses *a small number of token budgets $N=5$* and *short answers*, i.e. $|a_i|\le 10$. The per-question latency breakdown shows that **answer generation adds only 6.4% overhead relative to response generation**:
>
> | Generating a response y | Generating Answers a[1]..a[N] |
> |---|---|
> |  110.63 $\pm$63.80 s| **7.53$\pm$5.02s**|
>
> (b) **Faster Training Than Baselines**
>
> REO-RL trains faster than Vanilla RL and other RL methods with length rewards because *REO-RL encourages shorter responses.*
>
> | Method                          | Avg. Time per Training Step (s) | Avg. Length (Test) | Avg. Acc. (Test) |
> |---------------------------------|------------------------|-------------|-----------|
> | Vanilla RL                      | 1147.39                | 12609.9     | 71.0      |
> | **REO-RL (EXP)**                 | **877.36**                 | 6725.7      | 68.7      |
> | **REO-RL (Oracle)**                 | **873.26**                 | 7660.8      | 69.2      |
> | RL w. Length Group Norm. Reward | 938.58                 | 8961.3      | 69.8      |
> | RL w. Length-Harmonizing Reward | 903.93                 | 7956.8      | 70.4      |
> | RL w. Token Budget=4K           | 400.13                 | 2978.4      | 57.6      |
>
> Specially, RL w. Token Budget excessively shrinks length at the cost of severe accuracy drop,also having fast training speed.
>
> ### 2. Theoretical Cost Analysis
>
> Following [1] and [2], we analyze the computation cost for a single input $x$. Suppose the LRM is a transformer with $P$ parameters, KV size $D$ and GQA ratio $r$. Let $L_{in}$ be the length of the input, $L_{out}$ be the length of generated response $y$ and  $L_{ans}$ be the maximum lengths of answers generated from truncated responses. The computation in REO-RL consists of three parts:
>
> - Prefill for Input $x$: $2PL_{in}+2DL_{in}^2$
> - Decoding Response $y$: $2PL_{out}+r(2L_{in}+L_{out})L_{out}D$
> - Decoding Answers $a_1,\cdots,a_N$: $2PNL_{ans}+r(2L_{in}+2L_{out}+L_{ans})NL_{ans}D$
>
> **Why REO-RL is Efficient**:
>
> 1. **Negligible Answer Generation Cost.** The input length $L_{in}$ is small ($\le 500$) and the response length $L_{out}$ could span from 5K to 32K. In REO-RL, we use $N=5$ and $L_{ans}\le 10$. Therefore the computation cost is dominated by decoding $y$ since $L_{out}\gg NL_{answer}$.
>
> 2. **KV Cache Reuse:** The KV cache of $x$ and $y$ could be reused during answer generation. For more details on KV cache, please refer to [SGLang Blog](https://lmsys.org/blog/2024-01-17-sglang/).
>
> [1] Kinetics: Rethinking Test-Time Scaling Laws. https://arxiv.org/pdf/2506.05333
>
> [2] Large Language Monkeys: Scaling Inference Compute with Repeated Sampling. https://arxiv.org/abs/2407.21787
>
>
> ## [Q1] Further Details on Empirical Reasoning Efficiency Frontiers
>
> To ensure representative coverage of the best methods across varying token budgets, we run RL training with **3 classes of algorithms**, covering **8 different distinct algorithms** and **15 unique training configurations** in total. Each training configuration required substantial computational effort to ensure convergence, with individual run at least 2K and 4K GPU hours for each run in the 1.5B and 7B scales, respectively. For each run, a set of checkpoints are sampled for evaluation. In total, **186 and 218 models in 1.5B and 7B settings**, respectively, are evaluated for constructing the efficiency frontiers.
>
> For each model $\pi_{\theta_i}$, we first generate $32$ responses under a 32K token budget and then truncate each response under token budgets $\{32,64,\cdots,512,1K,2K,\cdots,32K\}$. We then generate answers from the truncated responses to compute the length-constrained rewards at different token budgets. Finally, the efficiency frontier for each benchmark is derived by selecting the **maximum accuracy across all models** for every token budget.
>
> The table below summarizes the algorithm classes, configurations, and model counts:
>
> | Algorithm Class         | Specific Algorithms and Training Configurations                                                               | # of Models (1.5B) | # of Models (7B) |
> |-------------------------|-------------------------------------------------------------------------------------|--------------------|------------------|
> | RL w. Token Budget      | Token Budget $\in \{512, 1K, 2K, 4K, 8K, 16K, 32K\}$                                    | 52                 | 118              |
> | RL w. Length Reward     | MRT; RL w. Length Group Norm. Reward ($\alpha\in\{0.1, 0.2\}$); RL w. Length-Harmonizing Reward ($\lambda \in \{1, 2\}$) | 83                 | 70               |
> | Preference Learning     | SimPO(Shortest, TOPS, DAST)        | 51                 | 30               |
> | Others | Base LRM, Vanilla RL | - | - |
>
>
> Further details about the algorithms and frontiers can be found in **Line 141-151** and **Appendix C**.

---

### Official Review · Reviewer_VFri · 2025-07-03

**Clarity:** 2
**Significance:** 2
**Originality:** 2
**Rating:** 4
**Confidence:** 2

**Summary:**

This paper tackles the problem of reasoning efficiency in Large Reasoning Models (LRMs). The authors observe that LRMs like DeepSeek-R1 often produce unnecessarily long reasoning traces - sometimes hundreds of tokens for simple problems. To address this, they introduce "reasoning efficiency frontiers" - empirical upper bounds showing the best accuracy achievable at each token budget. They then propose REO-RL, a reinforcement learning method that optimizes models to work well across multiple token budgets simultaneously. The key insight is using numerical integration to approximate performance across all budgets while only training on a small set (5 budgets).

**Questions:**

Have you tested on other domains like coding or general reasoning? Math problems might have specific properties that make this approach work well. Maybe presenting results such as puzzle or GPQA would be nice,w

**Ethical Concerns:**

["NO or VERY MINOR ethics concerns only"]

**Quality:**

2

**Strengths And Weaknesses:**

1.The paper identifies a real issue, the LRMs waste tokens. The "reasoning efficiency frontier" concept provides a concrete way to measure how far models are from optimal.
2.They test multiple fine-tuning approaches (RL with token budgets, length rewards, preference learning) to construct the frontiers. This gives a comprehensive view of current methods.
3.Only tested on math problems (AMC/AIME). It's unclear if this generalizes to other reasoning tasks like coding or general QA.
4. The "optimal" frontiers depend entirely on which methods they tried. If someone finds a better fine-tuning approach tomorrow, all the REG measurements become outdated.
5. The paper introduces many metrics (REG, length-constrained rewards, etc.) but it's not always clear which matter most for real applications.

Major concern is that the paper is not very well-written, not clear

---

> ### Author Rebuttal · Authors · 2025-07-31
>
> We sincerely thank the reviewer for the thoughtful feedback, which has helped us improve the clarity and scope of our work. Below, we address each concern in detail.
>
> ## Generalization to Other Tasks
>
> > "Only tested on math problems (AMC/AIME). It's unclear if this generalizes to other reasoning tasks like coding or general QA"
>
> To demonstrate the **generalizability of REG and REO-RL**, we conduct **cross-domain evaluations on coding tasks** using models trained solely on math data. Additionally, we apply the **REG framework and REO-RL** to coding by fine-tuning **Qwen3-4B** on DeepCoder data[1].
>
> **1. Cross-Domain Evaluation (Math -> Coding)**
>
> We evaluate models trained on math (as in the paper) on two coding benchmarks, CodeContests and LiveCodeBench. We construct **cross-domain efficiency frontiers** by truncating responses under different token budgets and allowing an extra 512-token budget for python code generation. We then compute REG based on the corss-domain efficiency frontiers. 8 responses are generated for evaluation for each method.
>
> Even when trained only on math, **REO-RL achieves strong generalization to coding**, significantly improving efficiency (lower REG) while maintaining accuracy.
>
> DeepSeek-R1-Distill-Qwen-1.5B Results:
>
> | Method | Acc. | Len.$\downarrow$ | REG $\downarrow$ |
> |----|-|-|-|
> | Vanilla RL | 13.55 | 15593.15 | 881.45 |
> | **REO-RL (Oracle)** |14.76 | 13064.89| **556.57** |
> | **REO-RL (Exp)** | 14.83 | 14047.32 | **510.56** |
> | RL w. Len Group Norm. Rew.  | 13.2 |14328	|996.29|
> | RL w. Len-Harmonizing Rew.|14.34|12670.71|646.61|
> |RL w. Token Budet=4K|13.35|6144.13|876.90|
>
> DeepSeek-R1-Distill-Qwen-7B Results:
> | Method | Acc. | Len.$\downarrow$ | REG$\downarrow$ |
> |----|-|-|-|
> |Vanilla RL|33.2|14722.48|849.27|
> |**REO-RL (Oracle)**|32.29|10246.18|**747.14**|
> |**REO-RL (Exp)**|31.98|9143.80|**789.20**|
> |RL w. Len Group Norm. Rew.|32.48|13183.96|839.00|
> |RL w. Len-Harmonizing Rew.|31.49|13240.02|1070.17|
> |RL w. Token Budet=4K|29.40|4968.69|1353.91|
>
>
> **2. Applying the REG & REO-RL to Coding (Qwen3-4B)**
>
> For frontier construction, due to time constraints, we run **RL with different token budgets** to estimate the efficiency frontier.
>
> Regarding REO-RL, we run REO-RL (Exp) using *$N=5$ exponentially spacing token budgets*. To obtain length-constrained rewards, we use the model to generate python codes from truncated responses with an addtional *generation budget of 512 tokens*. **REO-RL (Exp) reduces REG and response length by 51.14% and 32.49% on coding** while maintaining the overall accuracy.
>
>
> *Qwen3-4B Results on Coding:*
> | |Acc.| Length$\downarrow$|REG $\downarrow$|
> |-|-|-|-|
> |Base LRM (Qwen3-4B)|41.14|16850.23|514.86|
> |REO-RL (Exp)|41.03|11375.06|**251.55**|
> |RL w. Token Budget=8K|28.78|7059.58|2572.08|
> |RL w. Token Budget=16K|32.58|10443.31|1961.00|
>
>
> [1] DeepCoder: A Fully Open-Source 14B Coder at O3-mini Level. https://huggingface.co/agentica-org/DeepCoder-14B-Preview
>
> ## Potential Outdated Issue of REG
>
> > "The 'optimal' frontiers depend entirely on which methods they tried. If someone finds a better fine-tuning approach tomorrow, all the REG measurements become outdated. "
>
> We appreciate the reviewer’s insightful observation about the REG metric. Indeed, as with many performance metric in deep learning, REG’s frontiers may require updates as methods improve. We highlight two key points here:
> - Our large-scale benchmarking reveals a **fundamental gap that no existing method can fully match the estimated frontier**. This suggests that surpassing our current frontier would represent a meaningful advance in the field.
> - We present a comprehensive framework for estimating the efficiency frontiers. **New models or new methods can be seamlessly incorporated into the framework to update the frontiers and REG measurement**, keeping REG up-to-date.
>
> ## Clarity on Metrics (REG vs. Length-Constrained Rewards)
>
> >  "The paper introduces many metrics (REG, length-constrained rewards, etc.) but it's not always clear which matter most for real applications."
>
> We first would like to clarify the distinction:
> - **REG (Reasoning Efficiency Gap)** proposed as **a metric for evaluating reasoning efficiency**. It measures how close a model is to the optimal accuracy-length trade-off.
> - Length-constrained rewards are not metrics but used as rewards in REO-RL training, and for frontier estimation.
>
> Importantly, our results also suggest that **REG captures the trade-off between accuracy and generation length.** As shown in Table 1, methods that achieve low accuracy or long generation length tend to have high REG. In practice, REG can be used as an effective metric for measuring reasoning efficiency for any fine-tuned model.
>
> ## Improving Clarity
>
> We thank the reviewer for pointing out the clarity issue. We have taken the following steps to enhance the presentation:
>
> - **Training Efficiency Analysis**: We have added a detailed comparison with other methods, including a theoretical analysis of REO-RL’s computational cost, to explicitly demonstrate its training efficiency. (See our response to Reviewer 7ETN for details.)
> - **Frontier Construction Details**: We now include comprehensive statistics on the specific methods and models used for constructing the frontiers. (See our response to Reviewer 7ETN for details.)
> - **Improved Table Visualization**: As suggested by Reviewer s9q1, we have split Table 1 into sub-figures and sub-tables to improve readability and highlight key takeaways.
>
> We thank the reviewer again for the thoughtful comments. Please let us know if additional clarifications or revisions would be helpful.

---

### Official Review · Reviewer_s9q1 · 2025-07-06

**Clarity:** 3
**Significance:** 3
**Originality:** 3
**Rating:** 4
**Confidence:** 2

**Summary:**

This paper addresses the problem of reasoning efficiency in Large Reasoning Models (LRMs), which often generate excessively verbose reasoning traces despite their strong problem-solving capabilities. The authors introduce the concept of reasoning efficiency frontiers - empirical upper bounds on the optimal trade-off between accuracy and response length - and propose the Reasoning Efficiency Gap (REG) metric to quantify how far current methods are from optimal efficiency. The main contribution is REO-RL (Reasoning Efficiency Optimization with Reinforcement Learning), an algorithm that optimizes reasoning efficiency by targeting a sparse set of token budgets using numerical integration approximation. Experiments on mathematical reasoning benchmarks (AMC23, AIME24, AIME25) with DeepSeek-R1-Distill-Qwen models (1.5B and 7B) show that REO-RL reduces the efficiency gap by 74.5% and 64.2% respectively compared to vanilla RL.

**Questions:**

Please see the weaknesses. Addressing these issues will increase the clarity of the paper and help the readers better understand the main contribution of the paper.

**Ethical Concerns:**

["NO or VERY MINOR ethics concerns only"]

**Limitations:**

yes

**Quality:**

3

**Strengths And Weaknesses:**

Strengths
- Novel Framework and Metrics: The introduction of reasoning efficiency frontiers and the REG metric provides a principled way to evaluate and compare reasoning efficiency methods.
- Solid Experimental Design: The paper conducts comprehensive experiments with diverse fine-tuning approaches. The evaluation covers multiple baselines and includes proper ablation studies.
- Practical Algorithm: REO-RL cleverly approximates the computationally expensive full efficiency objective using numerical integration over strategically selected token budgets. The exponential spacing strategy achieves <1% approximation error with only 5 budgets, making the approach practical.
- Strong Empirical Results: The model finetuned with REO-RL demonstrates substantial REG reductions and competitive performance against frontier LRMs like Qwen3 and Claude Sonnet 3.7 in terms of reasoning conciseness.

Weaknesses
- Confusion in math notations: In section 4.1, when defining $a = Answer(π_\theta, x, y)$, the $L$ term is not shown as the input.  however, in equation (2), the $L$ term is shown as the input for $Answer()$. This may cause confusion while reading. Also, in definition 4.1 (equation (4)), as far as I am concerned, the reasoning efficiency frontier should be "a set" of best achievable reward $J$. In this case, maybe a notation like {$\hat{J}_{optimal}(D, \hat{\theta}, L) \mid L \in [1, Lmax]$} will help the reader better understand the technical details.
- Confusion in Table 1: Table 1 contains the main result of the paper. However, a lot of information (diverse baselines with different budgets, token length, accuracy across baselines, etc.) is contained in a single table. This could cause the readers unable to fully capture the main takeaways from the table. Maybe dividing the table into several sub-tables may be helpful (one table one main-takeaway).

---

> ### Author Rebuttal · Authors · 2025-07-31
>
> We thank the reviewer for the valuable review to our work. These valuable feedback could help us better improve the clarity of our work.
>
> **[W1] Confusion in Math Notations**
>
> We appreciate the reviewer’s helpful suggestions regarding notation clarity. We have accordingly revised the mathematical notations to ensure consistency and readability. Specifically, following the suggestion of the reviwer,
> - we use $a=\text{Answer}(\pi_\theta, x, y_{:L})$ to denote the answer for a truncated response $y_{:L}$, ensuring consistency with Eq.(2).
> - we use {$(L, \hat{J}_{optimal}(D, \hat{\theta}, L)))$ | $L\in[1, L{max}]$} to denote the reasoning efficiency frontier.
>
>
>
> **[W2] Confusion in Table 1**
>
> We agree with the reviewer that splitting Table 1 into more focused visualizations will enhance clarity. Specifically, in the revised version, we have included:
> - A figure comparing REG, showing that REO-RL consistently outperforms baselines in reasoning efficiency.
> - A line plot of accuracy vs. token budget, showing that Vanilla RL does not consistently improve accuracy across token bugets.
> - A visualization for accuracy, response length, and REG, illustrating how REG effectively balances trade-off between accuracy and response length.
>
> We believe these changes would greatly improve the clarity of our work. We deeply appreciate the reviewer’s time and valuable suggestions.

---

> > ### Comment · Reviewer_s9q1 · 2025-08-05
> >
> > I appreciate the responses from the authors. I think the enhanced clarify will be beneficial to the paper. I will keep my score.

---

### Comment · Area_Chair_Bzsp · 2025-08-04
**Gentle Reminder: Please Reply to Authors’ Responses (Only if Not Yet Done)**

Dear Reviewers,

As the discussion deadline approaches, may we kindly ask you to review the authors’ responses and post a constructive reply—unless you have already done so, in which case please kindly disregard this gentle reminder.

Your thoughtful engagement is deeply appreciated and essential to a fair and timely process. With sincere thanks for your continued dedication.

Area Chair

---

### Note · Authors · 2025-08-12

We sincerely thank the AC and all reviewers for their thoughtful feedback and constructive discussions during the rebuttal period. We appreciate the time and effort invested in evaluating our work. Below, we summarize the key points addressed in our rebuttal and the resulting improvements to the manuscript:

### **Key Contributions of Our Work**
1. **Efficiency Frontiers**: The first large-scale estimation of optimal reasoning efficiency.
2. **REG Metric**: A principled metric to quantify gaps to optimality.
3. **REO-RL (Exp)**: A training-efficient and practical algorithm, supported by empirical and theoretical analysis.

### **Major Rebuttal Discussions & Resolutions**
1. **Generalization Ability** (Reviewer Vfri,7ETN): We conducted cross-domain evaluations (e.g., coding tasks using math-trained models) and successfully applied REG + REO-RL to Qwen3-4B on coding task.
2. **Frontier Estimation Details** (Reviewer 7ETN,uvtF): Included details on training algorithms, configurations, and frontier construction.
3. **REO-RL Training Efficiency** (Reviewer 7ETN,uvtF): Provided empirical comparisons and theoretical analysis confirming its sample efficiency.
4. **Lack of Optimizing Answer Generation** (Revieewr uvtF): We resolved this concern via:
  - Theoretical analysis showing policy gradient includes both reasoning and answer optimization.
  - Additional ablation study confirming REO-RL as an effective approximation of the full objective.
  - Incorporated a missing technical detail highlighted by the reviewer.

### **Reviewer-Specific Responses & Improvements**
- **Reviewer s9q1**: We will improve Table 1’s clarity by splitting it into multiple figures (one per takeaway) and refine mathematical notations (Eq. (4)).
- **Reviewer Vfri**: Addressed concerns about REG’s potential outdated issue by clarifying that our framework can be recomputed to newer models, avoiding outdated issues.
- **Reviewer 7ETN**: Addressed concerns about scalability to ≥30B Models by applying REO-RL on Qwen3-32B, showing REO-RL’s effectiveness at larger scales.

We believe these revisions strengthen the paper and address all major concerns. Thank you again for your valuable feedback!

---

### Decision · Program_Chairs · 2025-09-17

**Decision:**

Accept (poster)

**Comment:**

Summary of the paper: This paper claims that current LRMs are far from optimal in the accuracy-vs-length trade-off and introduces the concept of “reasoning efficiency frontiers” to capture the empirically best accuracy achievable at every token budget. To quantify how far a model is from this frontier, the authors define the Reasoning Efficiency Gap (REG). They then propose REO-RL, a reinforcement-learning algorithm that minimizes REG by optimizing performance across a sparse set of token budgets whose rewards are integrated via numerical approximation. Experiments on AMC23, AIME24, and AIME25 with 1.5 B and 7 B DeepSeek-R1-Distill-Qwen models show that REO-RL reduces the efficiency gap by 74.5 % and 64.2 %, respectively, compared to vanilla RL, achieving competitive accuracy while being markedly more concise.

Strengths of the paper:
1. Useful framing: the “reasoning efficiency frontier” and REG metric give the community a concrete yardstick for evaluating conciseness vs. accuracy.
2. Practical algorithmic contribution: REO-RL’s use of sparse token budgets plus numerical integration keeps computational cost low (<1 % error with only 5 budgets).
3. Strong empirical results: large REG reductions across two model sizes, extensive baselines, ablations, and open-source code/data.

Weaknesses of the paper: I would like to express my heartfelt thanks to Reviewer uvtF—the most engaged and responsive reviewer. I am deeply grateful for the thorough and insightful review, as well as for the continuous discussions during the rebuttal period that helped us identify the shortcomings and refine the paper. I fully agree with the final comment: the current narrative feels disjointed and should be restructured. As for the other three reviewers, it seems to me that the authors have addressed most of the concerns. Please add the additional experiments and analyses, reorganize the paper, and correct the technical details to the camera-ready version of the paper.

Reasons for the decision:  This paper aims to rigorously quantify reasoning efficiency, provide a practical optimization method that yields measured gains, and is backed by solid experiments and open resources. In my opinion, while the frontier is empirical rather than theoretical and a structural concern was noted by Reviewer uvtF, these limitations are outweighed by the contributions; I therefore slightly lean toward acceptance. However, I would also comment to SAC on the concern that the Reviewer uvtF still has after the rebuttal.